# Sure-thing vs. probabilistic charitable giving: Experimental evidence on the role of individual differences in risky and ambiguous charitable decision-making

Philipp Schoenegger[1]*, Miguel Costa-Gomes[2]

1 School of Economics and Finance & School of Philosophical, Anthropological and Film Studies, University of St Andrews, St Andrews, Scotland, United Kingdom, 2 School of Economics and Finance, University of St Andrews, St Andrews, Scotland, United Kingdom

* ps234@st-andrews.ac.uk

**Data Availability Statement:** The data are stored on the OSF repository and freely available here: https://osf.io/w9gfu/.

## Abstract

Charities differ, among other things, alongside the likelihood that their interventions succeed and produce the desired outcomes and alongside the extent that such likelihood can even be articulated numerically. In this paper, we investigate what best explains charitable giving behaviour regarding charities that have interventions that will succeed with a quantifiable and high probability (sure-thing charities) and charities that have interventions that only have a small and hard to quantify probability of bringing about the desired end (probabilistic charities). We study individual differences in risk/ambiguity attitudes, empathy, numeracy, optimism, and donor type (warm glow vs. pure altruistic donor type) as potential predictors of this choice. We conduct a money incentivised, pre-registered experiment on Prolific on a representative UK sample (n = 1,506) to investigate participant choices (i) between these two types of charities and (ii) about one randomly selected charity. Overall, we find little to no evidence that individual differences predict choices regarding decisions about sure-thing and probabilistic charities, with the exception that a purely altruistic donor type predicts donations to probabilistic charities when participants were presented with a randomly selected charity in (ii). Conducting exploratory equivalence tests, we find that the data provide robust evidence in favour of the absence of an effect (or a negligibly small effect) where we fail to reject the null. This is corroborated by exploratory Bayesian analyses. We take this paper to be contributing to the literature on charitable giving via this comprehensive null-result in pursuit of contributing to a cumulative science.

## Introduction

Charitable giving has been the subject of much research across the social sciences. After all, substantial resources are spent every year on charitable ventures. For example, in 2019, US Americans donated just over 2% of GDP [1] to a total of up to 1.5 million registered non-profit organisations [2]. The vast majority of these decisions are made under conditions of risk (with

**Funding:** We hereby declare the following source of funding. One of the authors, Philipp Schoenegger, has received a research funding from the Forethought Foundation and the Centre for Effective Altruism (they do not provide grant numbers). The funders had no role in study design, data collection and analysis, decision to publish, or preparation of the manuscript. Links: https://www. forethought.org/ https://www. centreforeffectivealtruism.org/.

**Competing interests:** One of the authors, Philipp Schoenegger, has received research funding from the Forethought Foundation and the Centre for Effective Altruism. The charities used in this research are recommended by Giving What We Can, which is part of the Centre for Effective Altruism. Further, Philipp Schoenegger is a Global Priorities Fellow at the Forethought Foundation, which is part of the Centre for Effective Altruism. However, the funders had no role in study design, data collection and analysis, decision to publish, or preparation of the manuscript.

known outcome distributions) as well as ambiguity (with unknown outcome distributions). Plausibly, some charitable interventions have a high probability of providing the outcome promised; think of soup kitchens–where there is little uncertainty as to the intervention's ability to produce the outcome: If one donates a certain amount of money to a soup kitchen, a few additional litres of soup will probably be dispensed quite soon. However, other charitable interventions are more probabilistic in nature, i.e., they only have a (sometimes very) small chance of making an impact and some aspects of this calculation are difficult to articulate numerically; think of interventions aimed at reducing the risk of nuclear war. Here, the donor is confronted with substantial additional uncertainty over the likelihood of an event (like nuclear war inciting incidents) arising and ambiguity over whether the charity's interventions will have an impact on them should they arise: For all the donor knows, there might not be an incident risking nuclear war this year and even if there was, the charity might not be able to meaningfully influence the outcome. The former level of uncertainty can frequently be stated numerically, while the latter often cannot.

In this paper, we study actual donor decisions about these two kinds of charities: The first we call "sure-thing charities", where donations are technically still made under conditions of some risk, though their risk is quantifiable and comparatively minor. The second we call "probabilistic charities", where donations are also made under risk (e.g., relating to the likelihood of their event of concern arising), though other aspects of this choice are made under ambiguity (e.g., relating to the chance that an intervention will help address the issue that is itself extremely rare). When making choices between highly reliable and more uncertain options, variation in individual differences may drive heterogenous behaviour. In this paper, we specifically want to investigate whether individual differences in risk and/or ambiguity attitudes, empathy, numeracy, or donor type (warm glow vs. pure altruistic type) predict donation behaviour. Understanding why donors make the choices they do about these two types of charities has implications both for the academic literature and for fundraising efforts more directly.

There has been some previous work on the role of risk and ambiguity attitudes on pro-social behaviour in more abstract contexts. For example, Vives & FeldmanHall [3] find that ambiguity attitudes robustly predict pro-social behaviour, though that risk attitudes do not. Similarly, Chen & Zhong [4] find that uncertainty increases sharing behaviour in dictator games and reduces lying in dice games. In the context of other-regarding behaviour in binary dictator games, Haisley & Weber [5] find that dictators are less likely to choose an unfair distribution when the outcome allocation is dependent upon an ambiguous lottery rather than a risky one. For further similar research see [6, 7]. Additionally, Mesa-Vazquez, Rodriguez-Lara, & Urbano [8] find that when there is uncertainty about whether a dictator's choices are implemented, their actions are more generous.

A related but distinct literature looks at the effect of risk over donations themselves. Exley [9] find that in situations of risky donations where donors trade off personal payoffs and donations, donors give less, cf. also [10]. However, there is also some evidence for the opposite effect, namely that some types of uncertainty actually increase pro-social behaviour, cf. [11]. Further work investigates another type of uncertainty investigated in pro-social behaviour, namely uncertainty over whether any given recipient needs one's help. Engel & Goerg [12] find that uncertainty of this type "does not deter generosity" [12, p. 51]. When excluding selfish dictators, they find that in conditions of uncertainty donations are higher than under certainty. Further, Niehaus [13] suggests that because outcomes of charitable giving are usually not observed by the donors, decisions are primarily made on the perception of the outcome rather than the actual outcome. This would also be consistent with findings that show how many factors explain donor's reluctance to give to the most effective charities [14].

Our research builds on the literature on risk that has so far mostly employed directly controllable levels of risk in the lab. For example, in abstract game scenarios, risk can be controlled and stated precisely, for example by imposing a 50% chance of one's donation not being implemented, or by introducing a 5% chance that one's donation is matched. Crucially, our research is substantially different from the discussed literature primarily because we move the level of risk from precisely calculable interventions in the lab (as outlined above) to the actual charities themselves. While this introduces several design challenges, we argue that this step leads to an increased level of ecological validity of any potential finding. However, note that there is already a large literature on charitable giving generally that has a similar or higher level of external validity [15–17]. However, our paper's main contribution is the moving of our focus on risk and ambiguity to actual organisations and their interventions and away from aspects that can be controlled in the lab. Having risk and ambiguity at the level of actual charities is the level at which risk and ambiguity typically enter people's decision-making processes; rarely are we uncertain as to whether our donation will randomly increase when we donate (as in some experimental lab studies), but we are almost always acting under uncertainty about the charity's interventions that we consider donating to.

Additionally, there has also been a recently growing literature that adds individual differences in personality traits to standardly used predictors like risk preferences in a variety of economic contexts. For example, Knapp, Wuepper, & Finger [18] analyse the interrelations of risk preferences and personality as well as their predictive power in the context of farmer behaviour, finding that personality measures are among the best predictors of some of their economic behaviour. See also [19] on the role of the Big Five and [20] on the HEXACO model on economic behaviour, cf. also [21]. The role of personality has also been more directly studied in the specific context of charitable giving. Some find that personality traits can play a central role in charitable decision-making while others [22] see the effect as markedly smaller. Specifically, some have also found associations between empathetic character traits and donations as well as volunteering behaviour [23] while others have established a link between the Big Five personality trait of agreeableness and prosocial behaviour [24]. For further recent work on the relationship between personality traits and charitable giving, see [25, 26]. Our paper adds to this literature that aims to integrate personality traits into economics related studies and aims to advance our understanding of the role of individual difference measures to standard predictors of economic behaviour like risk and ambiguity attitudes concretely in the context of charitable giving.

In attempting the construction of an externally valid donation choice that allows us to capture the difference between sure-thing charities and probabilistic charities in actual charities that participants can donate money to, our main outcome variables of interest are (i) the choice between two real charities, one sure-thing charity and one probabilistic charity and (ii) the choice about one of those charities that has been randomly selected. Both of these charitable decision-making scenarios have strengths and weaknesses from an experimental design perspective, but we hope that they jointly allow us to better understand the role of individual differences in charitable decision-making scenarios like these. We outline the main weakness of (i) in the discussion section and argue that, overall, (ii) is a cleaner design.

In our experiment, each participant is first presented with a randomly selected pair of charities consisting of one charity of each type to control for accidental confounds relating the charity's context as each are presented with substantial additional accurate information to increase the naturalness of the choice. Participant choices with respect to this randomly selected charity pair then allows us to isolate and capture the element of probability between the two charity types. In the second part of this experiment, we study participant behaviour when they are shown only one randomly selected charity (either sure-thing or probabilistic), which more

narrowly captures the predictive value of individual differences on donation choices to charities of specific types. Overall, we find little to no evidence that individual differences in risk/ambiguity attitudes, numeracy, optimism, and donor type predict charitable giving behaviour. However, we do find that a purely altruistic donor type predicts donations to probabilistic charities when participants are either shown a sure-thing or a probabilistic charity. As such, we take this paper to be primarily reporting a null result.

## Hypotheses

In this paper, we are centrally interested in what best explains donations choices regarding sure-thing and probabilistic charities as we offer an account of moving the level of risk and ambiguity from risk over donations or outcomes in abstract lab environments to risk and ambiguity over actual charities' interventions, thus increasing the level of ecological validity (while clearly stating the design constraints and weaknesses that come with this move). To provide data on these questions, we pre-registered five concrete null hypotheses on the Open Science Framework (https://osf.io/w9gfu/).

First, we investigate whether donor choices can be explained by individual differences in risk and ambiguity attitudes. Previous work in domains such as stock market participation [27] and health-related field behaviours [28] has found that attitudes to risk and ambiguity can play significant roles. In the context of pro-social behaviour in game environments the results show that ambiguity aversion may play a role while risk aversion sometimes does not [3], though risk aversion has also been found to be "predictive for giving" [6, p. 95]. As such, we argue that given risk and ambiguity aversion have been shown to impact behaviour in many contexts including charitable giving, this makes it an a priori interesting relation to test. This hypothesis is also theoretically grounded, in that it might be the case that one's preference not to give to charities that have a low chance of making an impact might be driven by an individual's general risk aversion profile, or it might be that given the ambiguous nature of charitable interventions that it is only ambiguity aversion that impacts this choice. As such it is plausible that either (or both) of them may play a role in this choice. Our directional pre-registered prediction is that ambiguity and risk aversion predict donations towards sure-thing charities because those averse to risk and ambiguity may prefer charities that have clearly stated, and low-risk interventions as opposed to more ambiguous and risky ones. This is our first null hypothesis.

> Null Hypothesis #1: Ambiguity and risk attitudes do not predict choices between sure-thing and probabilistic charities.

Second, we investigate whether there are a few potential further individual difference measures that may help explain donor behaviour in the context studied here. Specifically, we will focus on numeracy, optimism, donor type (warm glow vs. pure altruistic type), and empathy. First, it may be the case that basic individual differences in numeracy explain a potential preference for probabilistic charities, primarily because decision-making in situations of small probabilities (and big potential payoffs) might be particularly difficult to comprehend for those less numerically versed. For example, previous research has found that those lower in numeracy were more insensitive to proportions of donation recipients [29] and that they showed higher susceptibility to changes in numeric presentation [30]. It may as such be the case that one's level of numeracy also meaningfully impacts behaviour in the context studied here as the probabilistic charities include interventions that have a small chance of making a large impact. Understanding these proportions plausibly requires a certain level of numeracy.

Further, one may also think that a general proclivity to optimism may bias individuals towards overestimating the success of probabilistic charitable interventions, or conversely that higher pessimism may explain a preference for sure-thing charities as those promise to have a reliable impact even in the worst-case scenario. This is corroborated by previous research that draws on the German socioeconomic panel, finding that optimism predicts charitable giving in some of their models [31]. Additionally, we investigate whether estimates of donor type, i.e. warm-glow vs. pure altruistic donor type, may predict behaviour too: For those on the warm-glow part of the spectrum, giving to a charity that has an immediate and reliable impact may have a higher chance of producing such a warm glow than giving to a charity that most likely will have no direct impact (or one that will not be observable for quite a while). Conversely, pure altruists would presumably primarily care about the perceived (expected) impact on social welfare and may as such be more likely to choose probabilistic charities on expected value grounds. Lastly, empathy has previously been shown to predict charitable giving frequency in a variety of contexts [32, 33]. It may thus be that empathy also explains choices in this study, for example if participants are preferring sure-thing charities due to their easier-to-relate-to interventions.

Our directional pre-registered predictions are that high numeracy, purely altruistic motives, and tendency towards optimism predict donations towards probabilistic charities. Conversely, our directional pre-registered predictions are that empathy and warm glow motives predict donations towards sure-thing charities. These predictions are encapsulated in our second null hypothesis.

Null Hypothesis #2: Individual differences in numeracy, optimism, empathy, and donor type do not predict choices between sure-thing and probabilistic charities.

One worry with the present design that might be raised is that despite the randomisation of charity pairs, any potential effect may still be driven by some level of contextual confounding present in the charity descriptions. To address this concern, we also have a condition in which participants make a choice between two 'context-free' charities, in which only fundamental information as to the underlying probabilistic aspects are preserved and the remainder of context is removed, though participants are informed about all aspects of the charity after their decision. This is our third null hypothesis.

Null Hypothesis #3: The factors predicting behaviour in NH1/NH2 do not predict behaviour in the blinded choice condition.

Fourth, it may be the case that if much of the potential preference for sure-thing charities is explainable by individual difference measures related to numeracy or risk attitudes, this might open up the possibility of shifting behaviour via informational interventions aimed at this. Specifically, we investigate how exposure to expected value reasoning affects donation behaviour. It might be the case that learning about or being made aware of expected value reasoning may meaningfully impact behaviour. Our directional pre-registered prediction is that those shown the expected value reasoning treatment text will be more likely to donate to probabilistic charities than those shown a made-up theory of decision-making (midontic decision-theory), which introduces participants to a nonsensical description of a theory of decision-making with the same directional recommendations, aimed at controlling for experimenter demand.

Null Hypothesis #4: The expected-value treatment does not impact behaviours in the sure-thing vs. probabilistic charity choice more than the control condition.

Lastly, one may also be interested in donation behaviour not between these two types of charities, but rather just in the context where potential donors are presented with one such charity. This may reduce the chance of additional confounds (like worrying that the design that presents two charities is artificial in its dichotomous presentation; after all, most naturalistic decisions are not decisions between two distinct choices). It also is overall a cleaner design that brings with it less drawbacks regarding interpretation of results. As such, we also investigate all our main hypotheses in the context where they are presented only with a single, randomly selected charity (equiprobable that it is a sure-thing or a probabilistic charity). This allows us to control for the type of charity and any potential confounds that this dichotomisation and variance in background knowledge might bring with it, as well as by presenting a cleaner design overall. As above, we study both the frequency and amount of giving with the same predictors (and directoinal predictions) as outlined in the previous sections. Our fifth null hypothesis below states this in detail and basically pools null hypotheses #1 and #2 into one single hypothesis.

Null Hypothesis #5: Ambiguity and risk attitudes (as well as individual differences in numeracy, optimism, empathy, and donor type) do not predict frequency and size of donations when presented with either a sure-thing or a probabilistic charity.

## Methods

### Participants and procedure

We pre-registered this project on the Open Science Framework (https://osf.io/w9gfu/), where we also deposited the full data set. For this study, we recruited a total of 1506 participants via Prolific (47.8% male) at a mean age of M = 45.09, SD = 15.49 from the United Kingdom. This sample is representative of the UK's population according to census data from the UK Office of National Statistics via Prolific's representative samples tool. This study received ethics approval from the University of St Andrews (approval code SA15429). Overall, participants receive £2 for participation and can earn up to £1.83 in addition (over three sets of tasks), depending on their choices in the experiment. We arrived at sample size of 1506 by conducting a pre-registered a priori power analysis in which we calculate that in order to obtain .95 power to detect an effect of the size $f^2 = .02$ (a standard measure of a small effect size which we picked as our smallest effect size of interest) at 5% alpha error probability in a linear multiple regression model, we would need to recruit a total of 652 participants. In order to account for those not wanting to donate (estimated at 33% as previous work has documented this rate to be between roughly 20% [34] and 40% [35]) and those failing comprehension questions, we aimed to recruit 1050 participants for the main condition and a further 450 for the remaining secondary conditions (for full specifications of all additional power analyses for further conditions and analyses, please see our pre-registration). In all results reported, we excluded a total of 261 participants who got more than one of the comprehension questions or attention checks incorrect. Our final exclusion criteria were: We excluded all participants who got more than comprehension/attention check incorrect. For all analyses in Final Choice, we also, in addition, excluded all those who indicated a donation that exceeded their earned endowment. This choice deviates from our pre-registered data exclusion plan. First, we did not pre-register that we would exclude data in Final Choice from participants who were stating a final donation higher than their earned endowment. As such, the final sample size used for all analyses is 1,245 participants.

In the first part of the experiment, participants provide fully informed consent, and then complete several individual difference measures relating to numeracy skills [36], tendency for optimism [37], and empathy [38]. For the completion of each these surveys, they are paid a

fixed endowment of £0.25 per survey (£0.75 in total) which is at their disposal for the remainder of the study. All participants then move onto the next part, where they are randomly selected into either the Main Choice condition (70% chance), the Expected-Value Treatment condition (20%), or the Context-Free Choice condition (10%). There, they decide to allocate some or all of their endowment to one of the two randomly chosen charities (one sure-thing and one probabilistic charity), or alternatively allocate none of it to either charity. After this choice, participants are asked to state their views on the impact of the charities that they had just seen on a 100-point scale. Further, they are also asked to estimate the average views of other participants on these charities. This task was incentivised by awarding participants an additional £0.10 if their estimates are within 5 percentage points of the actual average of judgements. This additional payment was paid out after the experiment ended and was not part of any donation decision in the experiment, because determination of estimation accuracy could not be done within the experiment itself. Following this task, they enter the second main section of the experiment, in which they complete two sets of choices that measure their risk and ambiguity attitudes, specifically a bomb risk elicitation task [39, 40] and an Ellsberg urn task. Across these two tasks, they can earn up to £0.98 in a second endowment, depending on their decisions and luck. All participants are then presented with one additional donation choice at Final Choice where they can decide to allocate all, some, or none of their earned endowment to one randomly selected charity (either a sure-thing or a probabilistic charity). For a complete overview of the experimental outline, see Fig 1. For an in-detail description of all steps and measures, see the section below.

## Measures

**Numeracy, empathy, optimism, and donor type measures.** In order to acquire measures of empathy, numeracy, and optimism, we use three validated, standardly used scales. Our numeracy measure is the 11-point numeracy scale [36] which includes three items on general numeracy and 7 items on risk numeracy specifically. We collect tendency towards optimism with the Life Orientation [37] which is a standard measure for optimism and pessimism. Further, we use the Basic Empathy Scale in Adults (BES-A) as our measure of empathy [38]. Participants are incentivised to complete these surveys by being rewarded with £0.25 for the full completion of each of the three surveys. We also collect self-reported donor type classification. To determine donor type (warm glow vs. pure altruist) we draw on Carpenter's [41] methodology that relies on a self-reported recollection of previous charitable behaviour, classifying donors as either warm glow, pure altruist, or other motivations.

**Risk and ambiguity attitudes.** We estimate risk and ambiguity attitudes via two distinct, fully incentivised tasks. In order to ensure participant comprehension, participants complete a trial run of the risk attitude measure, are shown the mechanism for the ambiguity mechanism in detail via an example, and answer comprehension questions on both measures. Specifically, we measure risk attitudes via the 'Bomb risk elicitation task' [39, 40], adopting the mechanism for Qualtrics directly from Nielsen [40]. In this task, participants are presented with a 5x5 matrix of boxes. They are informed that one box contains a bomb, and that all other boxes yield an additional income of 2 pence. Participants can select any numbers of boxes, but if the box containing a bomb is selected and they confirm that they have selected as many boxes as they wish, their income is zero. As Crosetto & Filippin [39] point out, this measure of risk attitudes has several upsides over other alternatives in that it "requires minimal numeracy skills [and] avoids truncation of the data" [39, p. 31]. Specifically, it "measures individual-level risk attitudes by a single parameter $k \in \{0, 1, \ldots, n\}$, the number of boxes collected" [42, p. 599]. Risk aversion is indicated by selecting fewer than 13 boxes and risk-seeking by selecting more

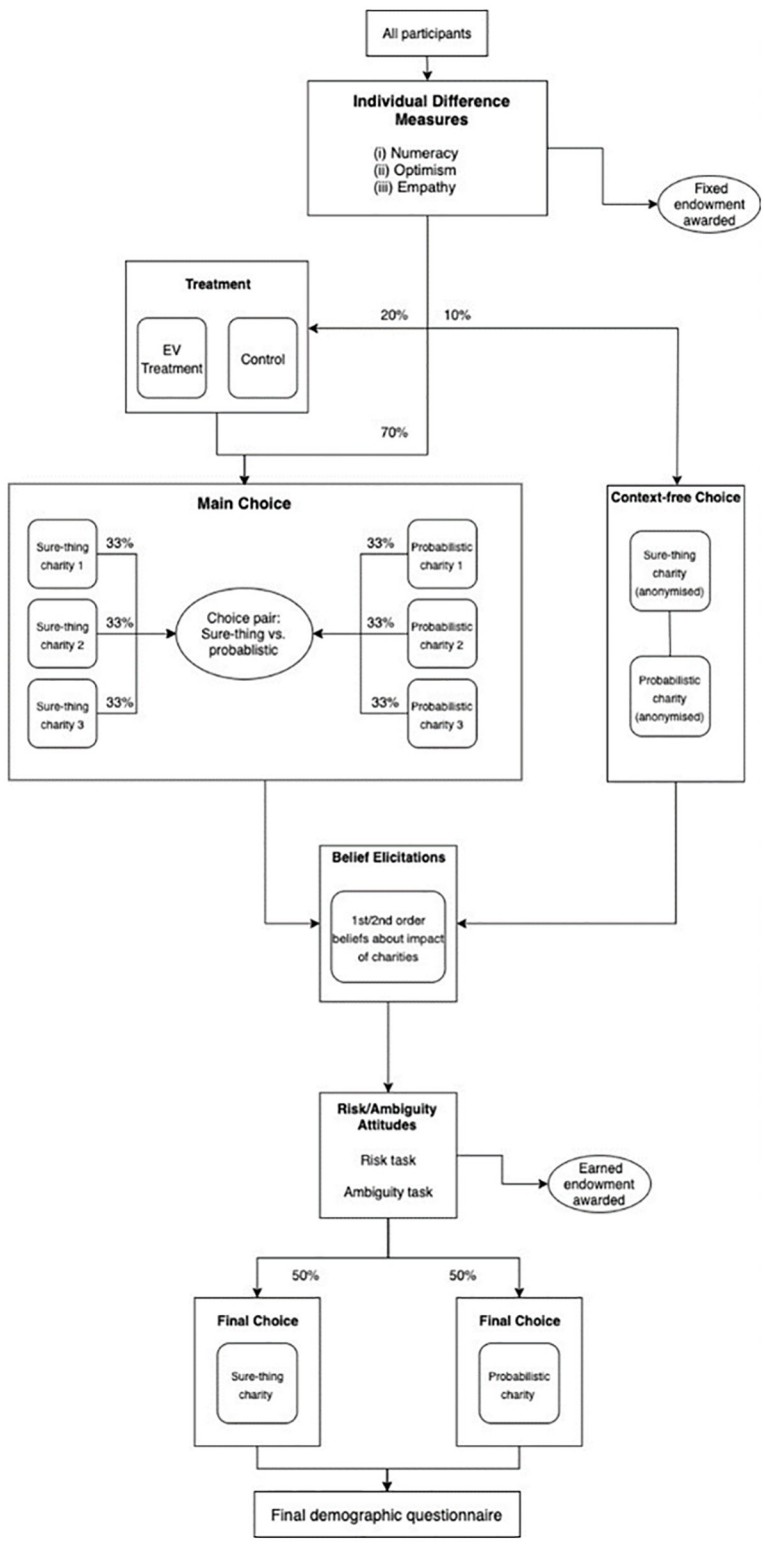

**Fig 1. Experimental outline.**

than 14. In effect, participants make choices between 26 lotteries ranging from choosing 0 boxes with a certain outcome of £0 to choosing all 25 boxes with a certain outcome of £0, while any number of boxes in between has a different chance to lead to additional income. The expected value of these lotteries is bow-shaped: playing lotteries with 13 and 14 boxes has the highest expected value, with choosing more or fewer boxes reducing expected value. As such, risk-neutral participants would choose 13 or 14 boxes, risk averse participants fewer than 13, and risk-seeking participants more than 14.

We measure ambiguity attitudes by recourse to standard Ellsberg urns: Participants are presented with two urns. They can bet on both the outcome of a risky urn containing 5 red and 5 black balls and on the outcome of an ambiguous urn containing an unknown quantity or both red and black balls. We then collect reservation prices for each urn by using a Becker-DeGroot-Marschak mechanism with random integers being uniformly drawn from the interval {1,2,. . .,25}. This design is adapted from Halevy [43]. The ambiguity premium is calculated by taking the difference between the ambiguous urn's reservation price and the risky urn's reservation price [43, p. 522–523] with negative outcomes indicating ambiguity aversion (and positive outcomes indicating ambiguity proneness).

**Demographic factors.** We also collect a number of additional demographic factors that are chosen as additional control variables. These are partially drawn from Bekkers & Wiepking [15] and Wiepking & Bekkers [44] who survey the literature and present a number of factors that predict charitable giving in a variety of contexts. Specifically, we collect data on age, gender, education, employment, religious affiliation and practice, marital status, children, and subjective financial well-being.

**Main choice and final choice outcomes.** We collect two main outcome variables for this study. These rely on participant choices when presented with the option to donate all, some, or none of their earned endowment to charity. Specifically, our two main outcomes are Main Choice (where participants make choices between a sure-thing and a probabilistic charity) and Final Choice (where participants make choices about a randomly selected charity). In both cases, we collect data on whether they donate at all, if they do, which charity they choose in Main Choice (participants can choose up to one charity), and how much they donate. In other words, participants can donate a non-zero amount to either charity or not donate at all (i.e., one cannot select a charity and choose to donate '0').

In Main Choice, each participant is shown exactly two charities, one sure-thing charity and one probabilistic charity. Each of these charities is drawn randomly from a pool of three charities (for a total of six individual charities and nine total charity pairs). This level of randomisation was chosen to avoid confounding this choice by the specifics of any given charity or any possible comparison effects between two specific charities. By randomising the pairs, we aim to isolate the factor of probability between the choices (sure-thing vs. probabilistic interventions) without having any potential result be influenced by the specific characteristics of any of these charities all the while keeping this choice as naturalistic as possible. Their choice here relates to null hypotheses #1 and #2.

In Final Choice, participants can again make a choice about their new endowment that they earned from the risk and ambiguity attitude tasks (ranging from £0 to £0.98, depending on their choices and luck). They are presented with one randomly selected charity (either a sure-thing charity or a probabilistic charity with equal likelihood) from the same pool as before and can again decide how much of their endowment to donate on this. We also ask participants before the uncertainty over their endowment in this section is resolved to state how much they would donate to this charity for different levels of endowments. This condition controls for a number of potential confounds in Main Choice which allows for it to answer the paper's

central question more directly and cleanly, though it being the last task of the experiment, we cannot rule out potential order effects. This choice relates to null hypothesis #5.

All six charities that are used in both Main Choice and Final Choice are recommended by GivingWhatWeCan in their 2021 charity recommendation titled "What are the best charities to donate to in 2021?" [45] to ensure that all of these charities are relatively similar in quality and reputation, and participants are informed about this fact prior to their choice. The three sure-thing charities are the SCI Foundation (deworming), GiveDirectly (direct cash transfers), and the Against Malaria Foundation (insecticide-treated bed nets). The three probabilistic charities are the Center for Health Security (epidemic/global health disaster prevention), Nuclear Threat Initiative (nuclear war prevention), and the Machine Intelligence Research Institute (artificial intelligence risk).

Our theoretical justification for this grouping is that all three sure-thing charities describe interventions that have a substantially high likelihood of success. All three interventions (deworming drugs, crash transfers, and bed nets) are administered by highly effective charities and have interventions that have been shown to have a high probability of being deployed successfully (i.e., that the cash transfer arrives, or the drug is administered). On the other side, all three probabilistic interventions (epidemic prevention, nuclear war prevention, artificial intelligence risk mitigation) are made under substantial ambiguity as to the probability distributions both of the intervention itself and the corresponding event occurring. These interventions are highly probabilistic in a way that cannot be resolved ex ante, making them qualitatively different than the sure-thing charities on a theoretical level, which is also recognised by the charities themselves and is as such part of their mission (after all, some of those risks are hard to quantify and even harder to address, but if such an intervention would be successful, it would bring with it substantially positive outcomes). For full treatment texts, see S7 Appendix.

However, we also provide an empirical justification for this grouping that supports the theoretical reasoning outlined above by conducting an auxiliary study. Note, however, that this study was conducted ex-post at the request of an anonymous reviewer and was not part of our original pre-registration. We conducted a post-hoc auxiliary study to empirically confirm our categorisation into sure-thing and probabilistic charities. This is to ensure that this distinction is not only theoretically grounded but also perceived as intended by the general public. We recruited a total of 101 participants on Prolific that had not participated in the main study, none of which failed the attention check. Participants were paid £0.75 for their participation. They were presented with all six charities and were asked to rate them on a scale from 0–10 on the likelihood that the charity's intervention succeeds (relating to uncertainty over its interventions) and on the quantifiability of the charity's intervention (relating to ambiguity). We also asked participants to rate the charities on their moral deservingness to keep the objective of this study relatively opaque.

We find strong support for the distinction between sure-thing charities and probabilistic charities on the basis of both uncertainty and ambiguity. See Table 1 for means, standard deviations, and medians of the uncertainty and ambiguity ratings for all six charities, with 0 indicating low probability that the charity's intervention will succeed and a low level of quantifiability of its interventions, and 10 indicating a high probability and quantifiability. In other words, the higher the scores, the less risky and the less ambiguous the charity's respective intervention is.

We find that the data behave as generally expected, with sure-thing charities receiving higher ratings about the likelihood that their interventions will succeed as well as higher ratings for the quantifiability of their interventions, and probabilistic charities receiving lower rating correspondingly. Adding a subject's scores for the individual charities of each bucket,

**Table 1. Probability-ratings for all six charities.**

| Uncertainty | | |
|---|---|---|
| | Mean (SD) | Median |
| Sure-Thing Charity 1 (SCI Foundation) | 7.19 (2.08) | 8 |
| Sure-Thing Charity 2 (GiveDirectly) | 5.92 (2.09) | 6 |
| Sure-Thing Charity 3 (Against Malaria Foundation) | 7.36 (1.83) | 8 |
| Probabilistic Charity 1 (Machine Intelligence Research Institute) | 4.15 (2.46) | 4 |
| Probabilistic Charity 2 (Nuclear Threat Initiative) fd | 4.09 (2.64) | 4 |
| Probabilistic Charity 3 (The Center for Health Security) | 4.30 (2.50) | 4 |
| Ambiguity | | |
| | Mean (SD) | Median |
| Sure-Thing Charity 1 (SCI Foundation) | 7.28 (1.97) | 8 |
| Sure-Thing Charity 2 (GiveDirectly) | 5.42 (2.49) | 5 |
| Sure-Thing Charity 3 (Against Malaria Foundation) | 7.24 (2.10) | 8 |
| Probabilistic Charity 1 (Machine Intelligence Research Institute) | 3.63 (2.72) | 3 |
| Probabilistic Charity 2 (Nuclear Threat Initiative) fd | 4.03 (2.73) | 4 |
| Probabilistic Charity 3 (The Center for Health Security) | 4.69 (2.40) | 4 |

*Notes*: Mean, Standard Deviation, and Median of risk and ambiguity ratings for all six charities.

we find that sure-thing charities are rated as having significantly higher probability interventions (M = 20.47, SD = 4.52) than the probabilistic charities (M = 13.96, SD = 5.83). This difference, 6.51, 95% CI [5.19, 7.82] was highly statistically significant, t(100) = 9.81, p < .001. The same picture emerges with regard to the quantifiability of the interventions, with the mean of the sum of the quantifiability scores of sure-thing charities (M = 19.83, SD = 4.60) being significantly higher than that of the probabilistic charities (M = 12.85, SD = 6.70), with the difference of 6.98, 95% CI [5.62, 8.34] also being statistically significant at t(100) = 10.17, p < .001. The effect sizes of these two differences, in Cohen's d, is d = .98 for the probability ratings and d = 1.01 for the quantifiability rankings. This provides strong support for our theoretically based distinction between sure-thing and probabilistic charities.

**Belief elicitations.** We elicit 1st and 2nd order beliefs regarding the charity's impact to investigate if participants who donate to one set of charities simply think that they are more impactful overall. Specifically, we ask participants to state their views on how impactful both charities they were presented with were on a percentage scale from 0 (Not impactful at all) to 100 (Extremely impactful). We also ask participants to estimate the average judgements regarding the impact of each charity as ascertained by the responses of all other participants. This is incentivised by paying those participants whose estimation is within 5 percentage points of the actual average an additional £0.10.

**Additional treatments.** We also had two additional treatments, an Expected Value Informational Treatment and a Context-Free Treatment. These were chosen to provide data that can speak to null hypothesis #4 and #3 respectively.

For our Expected Value Informational Treatment, we present participants with an introductory text of expected value reasoning. This is to test whether this simple information treatment can meaningfully shift donor behaviour. In order to control for experimenter demand effects, half of the participants in this condition are presented with a control text that introduces them to a made-up mathematical decision theory (midontic decision-theory). The control paragraphs introduce participants to this theory of decision-making via mathematically formulated but nonsensical arguments that are illustrated with the same example as the

treatment text. It recommends the same type of action, namely that taking risks can be worth it if the potential outcomes are good enough. This allows us to identify whether a given effect is due to experimenter demand or not by comparing a shift in frequency of donation to sure-thing/probabilistic charities between the expected-value treatment with the Main Choice and the made-up theory treatment (midontic decision-theory) and the Main Choice. See S7 Appendix for full texts. The choices here relate to null hypothesis #4.

In the Context-free condition, participants are given the option to choose between two charities that had all context (area of intervention, geographic focus, name, etc) removed. The only characteristics left are of the probabilistic nature of their interventions as well as some generic filler text. The participants are told that the full information would be made available to them right after they made their choice. See S7 Appendix for full texts. This condition is used to provide data about to null hypothesis #3.

However, because the number of people who made donations was unexpectedly small, both of these conditions did not have the power that we calculated prior to running this study to detect a meaningful effect. This means that results of these conditions are inconclusive. We still report the full pre-registered analyses in the appendix, see S4 Appendix, but do not discuss them in the main results and discussion sections due to this reason.

## Results

In Table 2, we summarise the general demographic variables and their frequencies with our sample. Generally, the sample shows high numeracy skills, with only 5.5% of participants not scoring at least 8 out of 10, and we observe a mean empathy score of $M = 76.54$, $SD = 9.75$ and mean optimism scores of $M = 33.08$, $SD = 7.20$. Further, according to Carpenter's [41] classification of donor types, we can classify 24.3% of participants as warm-glow donors, 11.9% as pure altruist donors, and the remaining 63.9% as being motivated by some other reason.

Our results for risk and ambiguity attitudes show a strong tendency towards risk aversion and a moderately strong tendency towards ambiguity aversion. Following [42] we measure individual-level risk attitudes by the number of boxes collected, k. Based on the cumulative distribution of participant choices, we find that 83.7% of participants exhibit risk aversion ($k < 13$), 8.9% of participants exhibit risk neutrality ($13 \ll k \ll 14$), and 7.4% exhibit risk-seeking behaviour ($k > 14$). Our ambiguity attitude measure is calculated by subtracting the ambiguous urn's reservation price from the risky urn's reservation price with negative outcomes indicating ambiguity aversion (and positive outcomes indicating ambiguity proneness). We find that 34.4% of participants exhibit ambiguity aversion, 46.6% of participants exhibit ambiguity neutrality, and the remaining 19.0% participants exhibit ambiguity-seeking attitudes. Additionally, we do not find a significant correlation between these two measures in the full sample with Pearson's $r = -.05$, $p = .079$.

Concerning the charitable giving outcomes in Main Choice, we find that across all conditions, 35.8% of participants make a donation. Of those making a donation, 84.7% donate to a sure-thing charity, and 15.3% donate to a probabilistic charity. The average donation of those who donate is 50.60 pence ($SD = 23.98$). For those donating to a sure-thing charity, the average donation is 51.50 pence ($SD = 23.83$), while for those donating to a probabilistic charity, it is 45.60 pence ($SD = 24.43$). This difference is not statistically significant, $t(305) = 1.557$, $p = .121$. We further find a strong relationship between choosing a type of charity and one's belief in its impact as well as its estimation of the general consensus of its impact by those making a donation. Specifically, we find that choosing to donate to a probabilistic charity stands in a strong positive point-biserial correlation to judging the probabilistic charity as impactful, $r_{pb} = .424$, $p < .001$, as well as estimating its average impact judgement, $r_{pb} = .346$, $p < .001$. Conversely, the

**Table 2. Demographics.**

| | n | % | | n | % |
|---|---|---|---|---|---|
| *Age* | | | *Religious Participation* | | |
| 18–28 | 247 | .198 | Yes | 112 | .090 |
| 29–38 | 216 | .174 | No | 1133 | .910 |
| 39–48 | 228 | .183 | | | |
| 49–58 | 228 | .183 | *Marriage Status* | | |
| 59 and above | 326 | .262 | Married | 586 | .471 |
| | | | Not married | 659 | .529 |
| *Gender* | | | | | |
| Male | 617 | .496 | *Children* | | |
| Female | 618 | .496 | Has children | 692 | .556 |
| Other | 10 | .008 | Does not have children | 553 | .444 |
| *Education* | | | *Financial Wellbeing* | | |
| High school | 498 | .400 | Finding it very difficult | 58 | .047 |
| Undergraduate | 446 | .358 | Finding it quite difficult | 149 | .120 |
| Graduate/Professional | 301 | .242 | Just about getting by | 408 | .328 |
| | | | Doing alright | 479 | .385 |
| *Religious Affiliation* | | | Living comfortably | 151 | .121 |
| No affiliation | 893 | .717 | | | |
| Protestantism | 162 | .130 | *Employment* | | |
| Catholicism | 120 | .096 | Unemployed | 174 | .140 |
| Islam | 34 | .027 | Out of the workforce | 223 | .179 |
| Hinduism | 15 | .012 | Part-time employment | 279 | .224 |
| Judaism | 10 | .008 | Full-time employment | 569 | .457 |
| Buddhism | 8 | .006 | | | |
| Sikhism | 3 | .002 | | | |

*Notes*: Demographics for the full sample after exclusion of 261 participants who failed more than one attention or comprehension check.

same choice also stands in a strong negative correlation with judging the sure-thing charity as impactful, $r_{pb}$ = -.314, p < .001, which is again mirrored in its estimated average impact judgement, $r_{pb}$ = -.267, p < .001. Looking at donation behaviour in Final Choice, we find that 29.1% of participants made a donation. 36.3% of those being presented with a sure-thing charity donated, compared to only 22.8% of those being shown a probabilistic charity. This difference is statistically significant, $\chi^2$ (1, N = 1177) = 25.66, p < .001. Of those making a donation, those being presented with a sure-thing charity donated an average of 26.58 pence (SD = 17.92) while those who were shown a probabilistic charity donated an average of 20.55 pence (SD = 16.55), a difference that is again statistically significant, t(340) = 3.45, p < .001.

First, we investigate general donation behaviour relating to Main Choice. The results presented in Table 3 speak to the central null hypotheses #1, #2, and #3. Model (1) reports the results for the Main Condition. The outcome variable is the type of charity conditional on a donation being made, with 0 being coded as the sure-thing charity and 1 as the probabilistic charity. The gender variable is coded 1 for female, all other categorical variables are coded 1 for the affirmative. As outlined above, the risk attitude measure is a discrete variable of the number of boxes opened, the ambiguity aversion is the result of the subtraction of the reservation prices. All other scales are the sum of the (re-reversed) individual items. As specified in our pre-registration, we report main regression results for binary outcomes using an OLS

**Table 3. Regression results for charitable giving behaviour in main choice predicting choice between sure-thing and probabilistic charities.**

|  | (1) |
|---|---|
| Risk Attitude | .000 (.005) |
| Ambiguity Aversion | -.003 (.005) |
| Numeracy | .004 (.024) |
| Empathy | < .001 (.003) |
| Optimism | -.004 (.003) |
| Donor Type |  |
| Warm-Glow | -.014 (.051) |
| Pure Altruism | -.062 (.064) |
| Donation (amount) | -.001 (.001) |
| Age | .003 (.002) |
| Gender | .013 (.050) |
| Education |  |
| Undergraduate degree | .044 (.051) |
| Postgraduate/Professional degree | -.020 (.055) |
| Religion |  |
| Protestantism | -.082 (.067) |
| Catholicism | .012 (.073) |
| Islam | -.101 (.122) |
| Judaism | -.163 (.379) |
| Buddhism | -.105 (.287) |
| Hinduism | .378 (.276) |
| Religious Participation | .041 (.089) |
| Marriage Status | .045 (.050) |
| Children | -.007 (.054) |
| Financial Wellbeing | .005 (.024) |
| Employment |  |
| Out of the workforce | -.154 (.097) |
| Part-time employment | .003 (.085) |
| Full-time employment | -.054 (.079) |
| $R^2$ | .064 |
| Sample size | 307 |

*Notes*: OLS regression reporting unstandardised coefficients and standard errors. Outcome variable is charity choice (0 = sure-thing charity, 1 = probabilistic charity). *$p < .1$, **$p < .05$, ***$p < .01$, ****$p < .001$

model and also for the corresponding logit model as a robustness check in Appendix Table 2 in S2 Appendix to check for sensitivity to functional form choice, where we find no impact of model choice. We made this choice primarily to enable clear and straightforward a priori power calculations, present more easily interpretable results, and because results are generally insensitive to model choice in most situations because the main reason against the use of linear models is that predicted probabilities may be greater than 1 or smaller than 0 which mainly occurs when the probabilities of the outcomes are extreme, not when they are in a roughly 80/20 distribution as is present in this paper [46–50]. We also report a robustness check in Appendix Table 3 in S2 Appendix where we report a regression with random effects at the stimulus level, finding that our null result is also robust to this model choice.

We find that none of the main variables nor the demographic control variables predict charity choice in the Main Condition. We can straightforwardly conclude from Model (1) that we do not have evidence to reject the null hypotheses #1 and #2 as none of the independent variables meaningfully predict donor behaviour. However, note that Model (1) includes the control variable of amount donated that was not pre-registered but suggested by a helpful anonymous reviewer. For the pre-registered regression model without this control with no difference in results, please see S1 Appendix.

Second, we investigate general donation behaviour in Final Choice where participants were either presented with a sure-thing or a probabilistic charity. In Table 4, we report two further regression models relevant to null hypothesis #5. These are not the pre-registered ones but instead include interaction terms that we did not pre-register. For specifications of the regression models as pre-registered, see Appendix Table 6 in S3 Appendix. Model (2) predicts frequency of donation and Model (3) predicts size of donation. For Model (2), not making a donation is coded as 0 and making a donation is coded as 1. The central variables are interaction terms, where we interact the main individual difference measures with the condition. Specifically, they are coded with 0 = sure-thing charity and 1 = probabilistic charity for all our main explanatory variables. In these analyses, participants from all conditions are included and are split only by which type of charity they were presented with at Final Choice; recall that each participant here was only presented with one randomly selected charity. For all analyses of Final Choice, we both apply the general exclusion criteria (excluding anyone who has answered more than one comprehension/attention task incorrectly) as well as the additional exclusion criterion where we exclude participants who indicated a donation that was higher than their earned endowment for Final Choice.

Here we find that while empathy predicts donation frequency and size overall, it is only the interaction term with pure altruistic donor type that statistically significantly predicts frequency of donation to probabilistic charities. This is in line with theoretical predictions that hold that pure altruists would be more likely to give to probabilistic charities as they primarily care about the (expected) impact on social welfare. Further, in our robustness check that reproduces Model (2) as a logit model (Appendix Table 5 in S3 Appendix), we find the same result. All other pre-registered variables (interacting with condition) do not predict donor behaviour.

Because we have found mostly null results in both Main Choice and Final Choice, we conduct the following exploratory analyses to further investigate the failure to reject the null hypotheses. Regarding the results from Main Choice, we conduct a series of equivalence tests for the multiple linear regression coefficients from Model (1) in Table 3 following Alter & Counsell [51] These allow us to make direct inferences about the absence of an effect or the presence of a negligibly small effect (i.e., an estimate of a precise zero) by providing two one-sided t-tests against an upper and a lower equivalence bound. In order for there to be sufficient evidence in favour of a negligible effect (i.e., evidence in favour of the absence of an effect or the presence of a negligibly small effect), both hypotheses have to be rejected at the .05 level. Because we did not pre-register the equivalence bounds, we report three plausible levels of the bounds in unstandardised coefficients at .01, .025, and .05 to show sensitivity of results to this choice. We report this for all our primary variables of interest from Main Choice that did not show significant effects.

These results suggest that we can be confident that the majority of our variables do not predict donor behaviour between sure-thing and probabilistic charities. Risk attitudes, empathy, and optimism show evidence in favour of a negligibly small effect at very tight equivalence bounds of (-).01, while ambiguity aversion shows this effect at (-).025, both relatively small levels. For numeracy, we can only conclude negligible effects at the (-).05 level. We remain unable

**Table 4. Regression results for charitable giving behaviour in final choice predicting frequency of giving and size of donation.**

|  | (2) | (3) |
|---|---|---|
| Risk Attitude | .002 (.005) | .124 (.120) |
| Ambiguity Aversion | -.002 (.004) | -.127 (.109) |
| Numeracy | .015 (.017) | -.193 (.453) |
| Empathy | .007**** (.002) | .212**** (.048) |
| Optimism | .003 (.003) | .062 (.071) |
| Donor Type |  |  |
| Warm-Glow | .061* (.047) | 2.170* (1.237) |
| Pure Altruism | .045 (.058) | -.144 (1.525) |
| Condition X Risk Attitude | .003 (.006) | .019 (.161) |
| Condition X Ambiguity Aversion | .002 (.005) | .085 (.143) |
| Condition X Numeracy | .006 (.018) | .275 (.482) |
| Condition X Empathy | -.002* (.002) | -.086 (.056) |
| Condition X Optimism | -.004 (.003) | -.033 (.088) |
| Donor Type |  |  |
| Condition X Warm-Glow | -.034 (.063) | -.993 (1.649) |
| Condition X Pure Altruism | .177** (.083) | 2.955 (2.176) |
| Age | .001 (.001) | .047 (.030) |
| Gender | -.050* (.030) | -.747 (.788) |
| Education |  |  |
| Undergraduate degree | -.050 (.031) | -1.129 (.800) |
| Postgraduate/Professional degree | .030 (.034) | -.095 (.897) |
| Religion |  |  |
| Protestantism | .071* (.043) | .153 (1.129) |
| Catholicism | .045 (.047) | 1.468 (1.239) |
| Islam | .173* (.142) | 6.606*** (2.317) |
| Judaism | .134 (.142) | 2.636 (3.730) |
| Buddhism | .159 (.160) | 4.994 (4.207) |
| Hinduism | .149 (.126) | 7.049** (3.301) |
| Sikhism | .046 (.260) | -2.343 (6.819) |
| Religious Participation | .015 (.054) | -.265 (1.406) |
| Marriage Status | -.025 (.031) | -.469 (.819) |
| Children | .032 (.033) | .697 (.877) |
| Financial Wellbeing | .023 (.015) | .398 (.384) |
| Employment |  |  |
| Out of the workforce | .017 (.055) | -.820 (1.442) |
| Part-time employment | -.019 (.047) | -2.211* (1.243) |
| Full-time employment | -.029 (.042) | -1.436 (1.107) |
| $R^2$ | .076 | .085 |
| Sample size | 1177 | 1177 |

*Notes*: OLS regressions reporting unstandardised coefficients and standard errors. Model (2) predicts frequency of donation and Model (3) predicts size of donation. Interaction terms interact the condition (0 = sure-thing charity, 1 = probabilistic charity) with the explanatory variables

*p < .1,

**p < .05,

***p < .01,

****p < .001

**Table 5. TOST for main choice coefficients.**

|  | -.01 | .01 | -.025 | .025 | -.05 | .05 |
|---|---|---|---|---|---|---|
| Risk Attitudes | 2.0** | 2.0** | 5.0**** | 5.0**** | 10.0**** | 10.0**** |
| Ambiguity Aversion | 1.4* | 2.6*** | 4.4**** | 5.6**** | 9.4**** | 10.6**** |
| Numeracy | .58 | .25 | 1.21 | .88 | 2.25** | 1.92** |
| Empathy | 3.32**** | 3.43**** | 8.32**** | 8.34**** | 16.66**** | 16.68**** |
| Optimism | 2** | 4.66**** | 7.0**** | 9.67**** | 15.33**** | 18.0**** |
| Warm Glow | -.08 | .47 | .22 | .76 | .71 | 1.25 |
| Pure Altruism | -.81 | 1.13 | -.58 | 1.36* | -.19 | 1.75** |

*Notes*: All t-test results for TOST procedures on a variety of lower and upper equivalence bounds (in unstandardized coefficients) from Model (1).

*p < .1,

**p < .05,

***p < .01,

****p < .001

to state clearly whether donor type has an effect on this choice as its results are insignificant both in the pre-registered regression analyses as well as the exploratory tests of equivalence. As such, we do not reach a conclusion as to the effect of donor type in Main Choice, but we can confidently state that the other explanatory variables of individual differences do not predict choices between sure-thing and probabilistic charities.

For the Final Choice analyses, we also conduct equivalence tests as above, see Table 6, with a focus on the interaction term coefficients from Model (2). We find that at the relatively tight (-).025 level, risk attitudes, ambiguity aversion, empathy, and optimism show a negligibly small effect in the frequency of donation model. Numeracy again shows a negligible effect at the (-).05 level and results regarding donor type (specifically warm glow) again do not allow for the conclusion of a negligible effect. We also report equivalence tests for the donation size in standardised coefficients in Appendix Table 10 in S5 Appendix, finding a similar pattern of results that shows numeracy and donor type as non-negligible effects while the remainder of variables show negligibly small effects.

These results provide us with a broad evidence base in favour of there not being an effect across many of our central explanatory variables. Also note that the results in Tables 5 and 6 are relatively robust to adjusting for multiple hypothesis testing. For example, if we adjusted our significance level to .007 following the Bonferroni method, we would still find that all significant effects remain significant at least at the (-).05 level.

**Table 6. TOST for final choice coefficients.**

|  | -.01 | .01 | -.025 | .025 | -.05 | .05 |
|---|---|---|---|---|---|---|
| Condition X Risk Attitudes | 2.17** | 1.17 | 4.67**** | 43.67**** | 8.83**** | 7.83**** |
| Condition X Ambiguity Aversion | 2.4*** | 1.6* | 5.4**** | 4.6**** | 10.4**** | 9.6**** |
| Condition X Numeracy | .89 | .22 | 1.72** | 1.06* | 3.11**** | 2.44*** |
| Condition X Empathy | 4**** | 6**** | 11.5**** | 13.5**** | 24**** | 26**** |
| Condition X Optimism | 2** | 4.67**** | 7**** | 9.67**** | 15.33**** | 18**** |
| Condition X Warm Glow | -.38 | .70 | -.14 | .94 | .25 | 1.33* |

*Notes*: All t-test results for TOST procedures on a variety of lower and upper equivalence bounds (in unstandardized coefficients) from Model (2).

*p < .1,

**p < .05,

***p < .01,

****p < .001

Further, we conduct exploratory Bayesian [52, 53] where we use Bayesian linear regression analyses that draw on Bayesian model averaging [54, 55]. First, in Table 7, we report results where we compare the individual models with one predictor each (the respective explanatory variable) against the null model with an intercept only. We report Bayes Factor Model Odds for the null for both Main Choice and Final Choice (frequency of donation as well as size of donation). For the analyses in Table 7, we use a uniform model prior.

The results in Table 7 indicate that for Main Choice, we have strong additional evidence in favour of a null effect of all variables of interest with Bayes factors of between 3.5 and 7.94. The picture is more complicated with regard to the interaction effects from Final Choice. There, we find that compared to models with ambiguity aversion or altruistic types individually, the null model is much more likely given the data. However, the other explanatory variables do not share this pattern. To further investigate the effect of our main variables on our outcomes of interest, we also report the model averaged coefficients that account for uncertainty over the estimates as well as uncertainty over model choice with all interaction effects and all control variables (age, gender, education, marital status, children, financial well-being, employment, religious affiliation, and religious participation). We report those coefficients as well as their 95% credible intervals that represent a weighted average that is weighted by the posterior probability of predictor inclusion. We use a uniform model prior and a JZS parameter prior with the default r scale of .354.

The results reported in Table 8 are in line with previous frequentist regression analyses, where we find that pure altruistic donor type predicts donations to probabilistic charities while most other interaction effects do not show any effects or only provide weak evidence in favour of them once all variables are entered into the model. As such, we conclude that the exploratory Bayesian analyses roughly corroborate our previous results.

## Discussion

Overall, we find little to no evidence in favour of rejecting our null hypotheses, and as such this paper primarily reports a negative result: First, we do not find evidence that risk and ambiguity attitudes robustly predict charitable decision-making between sure-thing and probabilistic charities (null hypothesis #1), and we also do not find evidence that individual differences

**Table 7. Bayes factor model odds for null model.**

|  | Main Choice | Final Choice (Freq.) | Final Choice (Size) |
|---|---|---|---|
| Risk Attitude | 7.46 |  |  |
| Ambiguity Aversion | 7.81 |  |  |
| Numeracy | 7.44 |  |  |
| Empathy | 7.89 |  |  |
| Optimism | 3.50 |  |  |
| Warm-Glow Type | 7.94 |  |  |
| Pure Altruistic Type | 5.94 |  |  |
| Condition X Risk Attitude |  | .04 | < .001 |
| Condition X Ambiguity Aversion |  | 13.41 | 7.57 |
| Condition X Numeracy |  | < .001 | < .001 |
| Condition X Empathy |  | < .001 | < .001 |
| Condition X Optimism |  | < .001 | < .001 |
| Condition X Warm Glow |  | 3.82 | 6.23 |
| Condition X Pure Altruism |  | 1.46 | 15.32 |

*Notes*: Bayes Factor Model Odds for the Null Model with Uniform Prior for both the explanatory variables for Main Choice and the interaction terms for Final Choice.

**Table 8. Bayesian linear regression coefficients and 95% credible intervals.**

|  | Final Choices (Freq.) | Final Choice (Size) |
| --- | --- | --- |
| Condition X Risk Attitude | .000 [-.001, .005] | .02 [-.03, .24] |
| Condition X Ambiguity Aversion | .000 [-.001, .005] | -.003 [-.008, .10] |
| Condition X Numeracy | -.01 [-.02, .000] | -.49 [-.89, .000] |
| Condition X Empathy | .000 [-.002, .001] | .001 [-.07, .06] |
| Condition X Optimism | -.001 [-.005, .000] | -.000 [-.03, .08] |
| Condition X Warm Glow | -.004 [-.01, .07] | .19 [-.25, 2.19] |
| Condition X Pure Altruism | .20 [.08, .32] | .90 [.000, 4.66] |

*Notes*: Model averaged coefficients and 95% credible intervals.

in numeracy, optimism, donor type, and empathy predict behaviour in this choice as well (null hypothesis #2). Further, we only find weak evidence predicting behaviour when participants are presented with either a sure-thing or a probabilistic charity, where we find that pure altruistic donor types significantly predict donation frequency to probabilistic charities (null hypothesis #5). Further, due to a lower than planned sample size, we are unable to conclusively evaluate the data about our no-context condition (null hypothesis #3) and with regard to the effect of the informational treatment introduced participants to expected-value reasoning (null hypothesis #4).

Importantly though, in conducting our exploratory analyses of equivalence tests and Bayesian analyses both in the context of Main Choice and Final Choice, we find that there is strong evidence in favour of negligibly small effects (or in favour of null effects) across a variety of plausibly set equivalence bounds and variable. Specifically, we find that for risk attitudes, empathy, and optimism, irrespective of the equivalence bound tested, their effects are negligibly small even at a delta level of an unstandardised coefficient at .01. For ambiguity aversion, the effect is negligible at .025, and for numeracy at .05. We do not have evidence in favour of a negligible effect for donor types. These patterns hold for both Main Choice and Final Choice with the caveat that in Final Choice, these tests were conducted with the interaction terms. We also conduct Bayesian analyses and find corroborating patterns of results. This provides evidence in favour of a null effect, something that the standard null-hypothesis testing technically cannot provide. In other words, these results suggest that risk attitudes, ambiguity attitudes, and individual differences in numeracy, empathy, and optimism have a negligible or no effect on charitable giving behaviour with regard to sure-thing and probabilistic charities (in both Main Choice and Final Choice designs). This is in itself an important finding that deserves to be recognised in the academic literature in an effort to combat file-drawer concerns and to expand our understanding of donation behaviour with regard to real charities.

Our results in Final Choice also provide us with some statistically significant effects. First, we find that empathy predicts both frequency and size of donation overall. This is in line with the previous literature on empathy and charitable giving [23, 32, 56]. Second, with regard to the central research question at hand, we find that in predicting charitable behaviour when participants are only presented with either a sure-thing or a probabilistic charity, we find that purely altruistic donor type predicts frequency of donation to probabilistic charities. This effect is directionally as predicted and is in line with theory in that pure altruists are more likely to be primarily concerned with an intervention's impact on social welfare and given the potentially high (expected) impact that probabilistic charities might have, the results provide evidence for this claim. This suggests that, at least compared to our analyses of the Main Choice, the data in Final Choice provide at least weak evidence in favour of our alternative hypotheses.

The results thus indicate that our hypothesised variables for explaining behaviour in the context of donating to sure-thing or probabilistic charities fail to meaningfully predict actual donor behaviour. However, these results do not by themselves allow for a concrete specification of why we fail to detect an effect. It may simply be that a different factor is at play when donors make these types of decisions and that individual differences in risk/ambiguity aversion do not impact decision-making at this level or that standard laboratory measures of risk/ambiguity attitudes does not directly translate into charitable decision-making in the context studied. It may also be that while this design was meant to provide a naturally occurring decision as is possible within the constraints of an online experiment, that a contraposition of two distinct charity types impacts decision-making such that participant choices are biased in some way. For example, this could be because of distinction bias, i.e., the bias describing how our preferences and choices may be substantially distinct in conditions where we evaluate options in separation or jointly [57–59]. We have some evidence in favour of this worry as at least some of the pre-registered factors predict directionally as expected in Final Choice, where only one charity was presented to participants. As such, it may be justified to put more interpretative emphasis on the results from Final Choice compared to Main Choice.

The results in Final Choice may also help reinforce the above claim in favour of the validity of this outcome measure and suggest that one reason for a failure to detect an effect in Main Choice may have been due to distinction bias related concerns, the bias that preferences and choices may differ between joint evaluation and separate evaluation modes [57–59]. On their own terms, these findings suggest that pure altruists are more likely to donate to probabilistic charities compared to other donor types (like warm glow donors).

Further, it may be that participants did not pick up on the distinction between the two charities and that our outcome variable was faulty in some way. If this was the case, this would threaten the validity of all results presented here. There are several reasons to believe that this is not the case. First, the qualitative comments at the end of this study often explicitly raise concerns of probability, risk, and chance with regard to the participants' charity choice, suggesting that they did pick up on the central difference. Second, we find that participants who favour probabilistic charities also judge them as more impactful and also estimate that the average judgements of other participants as higher than those who favour sure-thing charities. Lastly, we conducted an auxiliary study in which a new set of participants rated the charities among other things on the dimension of risk. There, we find that sure-thing charities are rated as significantly less risky and less ambiguous than probabilistic charities ($p < .001$). This gives us some prima facie evidence in favour of the validity of our outcome variable, though they are not (and indeed cannot be) conclusive. However, it is worth pointing out that this auxiliary study cannot help us rule out whether the two groups of charities also differed along other dimensions in the perceptions of participants.

It is also worth pointing out that while most of the previous literature discussed above uses manipulations of risk and ambiguity that are more tightly controlled (i.e. by introducing risk over donations controlled by the experimenter), this study attempted to provide a more naturalistic and thus externally valid picture of actual risky and ambiguous charitable giving, and failing to provide evidence in favour of this may also be partly explained by the general difficulty that highly abstract concepts in perfectly controlled laboratory games have when they are moved towards more externally valid contexts, cf. [60]. As such, failing to provide further evidence in favour of the impact of individual differences in, say, risk attitudes, on charitable decision making may be best understood as being, at least in part, explainable by a failure of experimental measures to generalise above and beyond their original context (as the outcome variables in this study were actual charities and not, like similar literature before, allocation decisions in dictator games where risk can be varied between conditions quite precisely). This in itself is not a bad

thing, and we take our finding to contribute to a cumulative science that aims to establish which measures generalise in what contexts and situations and where such generalisations fail (as is the case in this paper). Additionally, it is important to point that much of the literature that this paper builds on is based on fundamentally different experimental designs. As Frey et al. [61] have already shown, behavioural elicitation methods for risk preferences often show low correlations between each other, making it not as straight-forward as presented previously to assume a similar underlying relation of different elicitation methods. While we argue that this is not primarily a weakness of the currently, we see it as general issue with the field overall, suggesting that more specific cognitive models and theories that build stronger connections between elicitation methods, constructs, and cognitive processes influencing behaviour would be sorely needed. We thank an anonymous referee for pressing us on this point.

Moreover, there is an additional reason to significantly favour the results of Final Choice over those of Main Choice throughout the interpretation of this paper's data. This is because the design of Main Choice has some inherent flaws that make interpretation of results difficult. We thank an anonymous referee for pressing us on this point. This is because when participants make choices between two charities, they can make choices between compositions that they are indifferent between. For example, any one donor may be indifferent between a donation of £0.25 to a sure-thing charity and a donation of £0.10 to a probabilistic charity. In this scenario, the donor would choose randomly between the two. This in itself makes it difficult to cleanly identify a preference for one charity type or the other in the design that we have pre-registered and analysed in this paper. Based on this reason, we argue that the results from Final Choice should be given greater weight compared to Main Choice throughout this paper.

This concern is even more relevant if one considers that it may be the case that donors could be indifferent between donating a positive amount to a sure-thing charity and no donation to a probabilistic charity. As before, if they are indifferent between the two and choose randomly, this inhibits straightforward interpretation of our results. This is because we exclude all participants who choose not to donate (to understand their behaviour between these two options). Further, because it is quite plausible that, given the distribution of donation choices (most people who donate donate to a sure-thing charity), we may be excluding significantly more people favouring probabilistic charities but not sure-thing charities from those who are indifferent between a non-zero donation to a sure-thing charity and a zero donation to a probabilistic (and thus choose randomly between the two).

Some of these concerns have been addressed ex-post in this paper, for example by controlling for amount donated in the analyses of Main Choice that we did not initially pre-register. While this goes some way towards addressing this concern, we argue that some fundamental design constraints of the set-up that we chose remain. As such, we have put a higher emphasis on our analyses of Final Choice (which do not fall prey to the same structural challenges) throughout this paper and argue that one ought to be generally cautious in interpreting the results of Main Choice. However, given that it was pre-registered, we continue to report it fully (and where we deviate from the pre-registered protocol, we document this in detail and provide the original analyses in the appendix). We hope that this discussion, highlighting these issues in detail, properly contextualises the results for readers.

Overall, given that that we find little or no evidence in favour of rejecting our null hypotheses as well as provide evidence in favour of a null effect across both of our main hypotheses, we take our paper to primarily report a negative result. Given our relatively high level of power that was calculated a priori, and the fact that all analysis steps were pre-registered (and where we deviated from the pre-registered protocol, we documented this and provided the pre-registered analyses in the appendix), and because our equivalence tests provided strong evidence in favour of negligibly small results, we can be relatively confident in this result (at least in the

data provided by Final Choice). Because it is important for any science aiming to be cumulative and reproducible, it is imperative that null results like these are communicated openly and clearly, and we take our paper to be doing precisely that.

## Limitations

One potential limitation of this study is that our final sample sizes used for analyses of Main Choice were lower than those specified in the a priori power analyses. While we recruited the exact number that we preregistered, this reduction in sample size was primarily because a lower percentage of participants decided to donate to any charity than expected, i.e., in our data set 35.8% of participants choose to make a donation, while we expected about two-thirds of participants to make a donation [34, 35] However, because our power analyses were conducted with the highly set goal of having .95 power to detect the smallest effect size of interest at $f^2 = .02$, we were only slightly below the sample size required for the more conventional level of power at .8 (which would have required 395 participants), which should alleviate these concerns at least in part. Additionally, these sample size concerns do not apply to any analyses conducted with regard to our Final Choice data due to the difference in design. As such, while these worries ought to be brought to the attention of readers, we do think that they are ultimately manageable in scope and do not take away from the findings of this paper.

A second potential limitation is the use of a web-based sample from Prolific. While we have tried to counterbalance this concern by relying on a representative sample (at least along the dimensions of age, sex, and ethnicity), there may still be some dimension along which our sample is not representative of the population as a whole. This, in turn, means that that there may be some external validity worries inherent in using this sample, which may impact generalisability of our results.

## Conclusion

In this paper, we have attempted to investigate what best explains individual charitable giving behaviour in situations where donors are presented with highly reliable charities (sure-thing charities) and significantly riskier charities (probabilistic charities). We have failed to reject most of our null hypotheses in the context of donor behaviour when choosing between the types of charities and can only offer mixed results with regard to donor choices when they are presented with either a sure-thing or a probabilistic charity. There, we find that donor type meaningfully predicts frequency of donation to probabilistic charities. Exploratory results of equivalence tests, however, provide positive evidence in favour of the absence of an effect (or the presence of a negligibly small effect) across most of our main explanatory variables. Overall, we take this study to produce a robustly negative result in that our hypothesised variables failed to predict donor behaviour, while also producing positive evidence in favour of the claim that individual differences in risk/ambiguity attitudes and individual differences in numeracy, optimism, and empathy do not predict choices with regard to these two types of charities.

## Supporting information

**S1 Appendix. Preregistered model specification for model (1).**
(PDF)

**S2 Appendix. Robustness checks for main choice.**
(PDF)

**S3 Appendix. Robustness checks for final choice.**
(PDF)

**S4 Appendix. Additional hypotheses tests with inconclusive results.**
(PDF)

**S5 Appendix. Additional equivalence tests.**
(PDF)

**S6 Appendix. Additional regression for main choice.**
(PDF)

**S7 Appendix. Experimental texts.**
(PDF)

**S1 File.**
(PDF)

**S2 File.**
(DOCX)

**S1 Data.**
(CSV)

**S2 Data.**
(CSV)

## Acknowledgments

We thank for helpful comments, suggestions, and support Kirby Nielsen, Julian Jamison, Andreas Mogensen, Maximilian Maier, Joshua Lewis, Ben Grodeck, Theron Pummer, Ruru Hoong, and Thomas Rowe, as well as participants at the 2021 Early Career Conference Programme at the Global Priorities Institute, University of Oxford.

## Author Contributions

**Conceptualization:** Philipp Schoenegger, Miguel Costa-Gomes.

**Data curation:** Philipp Schoenegger.

**Formal analysis:** Philipp Schoenegger, Miguel Costa-Gomes.

**Funding acquisition:** Philipp Schoenegger.

**Investigation:** Philipp Schoenegger, Miguel Costa-Gomes.

**Methodology:** Philipp Schoenegger, Miguel Costa-Gomes.

**Project administration:** Philipp Schoenegger, Miguel Costa-Gomes.

**Resources:** Philipp Schoenegger.

**Software:** Philipp Schoenegger.

**Supervision:** Miguel Costa-Gomes.

**Validation:** Philipp Schoenegger.

**Visualization:** Philipp Schoenegger.

**Writing – original draft:** Philipp Schoenegger.

**Writing – review & editing:** Philipp Schoenegger, Miguel Costa-Gomes.

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
