## [Decision Letter · Decision Letter 0]

17 Feb 2022

PONE-D-21-35635Sure-Thing vs. Probabilistic Charitable Giving: Experimental Evidence On the Role of Individual Differences in Risky and Ambiguous Charitable Decision-MakingPLOS ONE

Dear Dr. Schoenegger,

Thank you for submitting your manuscript to PLOS ONE. After careful consideration, we feel that it has merit but does not fully meet PLOS ONE’s publication criteria as it currently stands. Therefore, we invite you to submit a revised version of the manuscript that addresses the points raised during the review process.

We look forward to receiving your revised manuscript.

Kind regards,

Junhuan Zhang, PhD

Academic Editor

PLOS ONE

Journal Requirements:

Reviewers' comments:

Reviewer's Responses to Questions

**Comments to the Author**

1. Is the manuscript technically sound, and do the data support the conclusions?

Reviewer #1: Partly

Reviewer #2: Partly

2. Has the statistical analysis been performed appropriately and rigorously? 

Reviewer #1: Yes

Reviewer #2: Yes

3. Have the authors made all data underlying the findings in their manuscript fully available?

Reviewer #1: No

Reviewer #2: Yes

4. Is the manuscript presented in an intelligible fashion and written in standard English?

Reviewer #1: Yes

Reviewer #2: Yes

5. Review Comments to the Author

Reviewer #1: Summary and Overall Evaluation

In this article, the authors examined the effect of several individual difference measures, among them, risk and ambiguity preferences, numeracy, empathy, optimism, and donation motivation. The main dependent variable is if and if then how much participants were willing to donate from their experimental endowment to one of two different types of charities, which the authors call sure-thing or probabilistic charities. Another dependent variable was the donation to an individual charity (either sure-thing or probabilistic) at a later stage of the experiment. In addition, the authors examined whether an intervention, namely advertising the EV-maximizing principle as a decision criterion, has an effect on donation choices and whether presenting anonymous and context-free donations rather than real-world charities with concrete description of their cause lead to different behavior than the main condition. As results, none of the individual difference measures significantly affected the choice of donating to a sure-thing or probabilistic charity, and only empathy and donation type affected overall donation in the choice to donate to an individual charity in some models.

The paper is clearly written and reports all methods, hypotheses and analyses in an understandable fashion. I appreciate that the study was pre-registered and I think the general research question is important and interesting. Moreover, the individual difference measures are all taken from previous research and seemed to be reasonable choices to measure the respective constructs. However, I am not fully convinced of the classification into sure-thing and probabilistic charity, of the theoretical underpinning of the hypotheses, and the power of the study. I will discuss each of these issues in more detail in the next section.

Major Comments

1. The main dependent variable is the choice between so-called sure-thing and probabilistic charities. In total there were six charities, three classified as sure-thing and three classified as probabilistic. This classification is rather ad-hoc and neither rooted in empirical evidence, nor based on principled theoretical arguments. Whereas I might agree from reading the description that this classification could be justified, the current study did not establish that all or even the majority of participants perceived the charities as assumed in the classification. Instead, there is a lot of information in the descriptions of the charities and participants might focus on different information than impact probability. Second, participants could perceive other information about the charity as varying in riskiness. For example, the direct money transfer could be perceived as risky with respect to the ultimate use of the donated money. Finally, the charities in the different conditions could also differ on other dimensions than riskiness. For example, the sure-outcome charities all have a direct impact on individual people, whereas the probabilistic charities have a more indirect impact on a large amount of people. To be fair, the authors mention concerns about the classification of charities into sure-thing and probabilistic in the discussion. However, the provided evidence in favor of their classification are not convincing as they do not directly speak towards a difference in perceived riskiness between the two types of charity. Instead, a straight-forward way to examine this question empirically would be to ask participants either from the old participant pool if available or a new one, how much they rate the respective charities on several dimensions including the riskiness of the project.

2. Since there were different charities administered in the stimulus material, it would make sense to estimate regressions with stimulus random effects, instead of OLS. Other models that cluster errors on the stimulus level might also be possible. Similar to the logistic, this could be done as a robustness test.

3. More effort should be spent on the theory of why the examined individual differences should be correlated with the choice between the sure-thing and the probabilistic charity. In particular, why should risk preference be related to this choice, if—as mentioned by the authors—previous literature (Vives & FeldmanHall, 2018) did not find a connection between risk-preference and prosocial behavior, that the exact probabilities of success of a charity are most likely ambiguous and that risk- and ambiguity preferences are not strongly correlated? Similarly, the connection between optimism or numeracy and the main DV is not build on theoretical considerations nor on previous findings that suggest such a connection. A strong foundation of the hypotheses would considerably strengthen the importance of the article.

4. There is some uncertainty about the power of the regression results as the pre-registered power analyses were based on the assumption that 2/3 donated, but the true number of donations was only 1/3. I appreciate that this is mentioned in the discussion, but I think this aspect deserves a bit more attention. In particular the EV-max intervention and the context-free manipulations with a much smaller sample size than the main dependent variable, might not have enough power to draw definite conclusions. Instead, it might make sense to put these analyses in an appendix and label these analyses as inconclusive. For the main DV, I appreciate that the authors calculate equivalent tests to examine the measurement uncertainty around the true effect. In my view, a better or at least complementary approach would be to calculate Bayes factors to evaluate the evidence for the Null (see Jarosz & Wiley, 2014; Rouder & Morey, 2012). Bayes Factors could clearly state whether enough evidence has been collected to conclude that there is no effect of an individual difference measure on the choice between the sure-thing or the probabilistic charity. In case that the evidence is inconclusive, it might make sense to collect more data.

Minor Comments

I would suggest to describe the hypotheses in the introduction in terms of the alternative hypothesis. Presenting hypotheses as the Null makes the text wordier and more complex than necessary.

On page 5 it is stated that:

“Our research builds on this literature but is importantly different, primarily because of our focus on donations to actual charities and not on pro-social behaviour in abstract games.”

And on page 6:

“Specifically, we focus on donation behaviour between real world charities that are made with an earned endowment, in contrast to abstract laboratory game pro-social decisions and hypothetical choices.”

These comments suggest that there is basically no research about charitable giving and all research about social preferences is only done in abstract experimental paradigms. I think this is a wrong impression and literature about charitable giving should be properly cited in these situations. Possible literature and review articles of which some are also cited in the text are Bekkers and Wiepking (2011), Karlan and List (2007), and Vesterlund and Sonnevi (2007).

References

Bekkers, R., & Wiepking, P. (2011). A literature review of empirical studies of philanthropy: Eight mechanisms that drive charitable giving. Nonprofit and voluntary sector quarterly, 40(5), 924-973.

Jarosz, A. F., & Wiley, J. (2014). What are the odds? A practical guide to computing and reporting Bayes factors. The Journal of Problem Solving, 7(1), 2.

Karlan, D., & List, J. A. (2007). Does price matter in charitable giving? Evidence from a large-scale natural field experiment. American Economic Review, 97(5), 1774-1793.

Rouder, J. N., & Morey, R. D. (2012). Default Bayes factors for model selection in regression. Multivariate Behavioral Research, 47(6), 877-903.

Vesterlund, L., & Sonnevi, G. (2006). 24. Why Do People Give?. In The nonprofit sector (pp. 568-588). Yale University Press.

Vives, M. L., & FeldmanHall, O. (2018). Tolerance to ambiguous uncertainty predicts prosocial behavior. Nature communications, 9(1), 1-9.

Reviewer #2: While I was reading the paper, I found the research questions interesting and important. I liked it a lot. However, when I read though the experiment design, I found, unfortunately, the design in the Main choice and context-free choice is not appropriate to answer the research questions. The concern is that the current design asks the participants to make two choices: which type of charity to donate to and how much to donate, while these two decisions are correlated with each other. Consequently, the experiment is not well controlled.

To make this more clear, I’ll first describe what is an ideal design and then point out the issues about the current design. To answer the research questions or test the hypotheses, we can either test whether they prefer to donate to sure-thing charity or probabilistic charity or test how much they would like to donate given sure-thing charity or probabilistic charity. In the former case, we need to control for the donation level, i.e. if they donate, they donate the same amount of money. In the latter case, we randomly control for the charity type. The latter case is of course the Final Choice design in the current paper which I fully agree is the correct way.

The issue with the current design is that, theoretically, there should be a point where one is indifferent between donate a certain amount of money under sure-thing charity and another amount under probabilistic charity. So what we watched in current experiment is just one of the two choices they are indifferent from. Therefore, it is hard to tell from their preference for the charity type.

To extend the above point a bit more and perhaps it helps to make it more clear, suppose there are some amount of people who would donate zero no matter what type of charity they are assigned or chose in the Main choice (this happens I guess for sure because there are only 35.8% of participants made a donation). For these people, economically, their choice of the charity type is invalid as the cost is zero (they will not donate anyway). This is an extreme case.

Now return to the paper, to save it, first, I think the Final Choice design is good, the authors may want to rely more on the data generate from the Final Choice. But at the same time take it in mind that this is last task which may suffer from order or experimenter demand effect. Second, if the authors really want to use the data from the Main Choice, they should at least control for the donate amount in the regression. Though I still think this is not valid enough to test the hypotheses.

In table 4, to test the hypothesis using the Final Choice data, they should add in interactive term between treatment variable (sure-thing or probabilistic) and the major explanatory variables to test the difference in difference, that is whether the major explanatory variables can explain the donation difference under sure-thing or probabilistic charity (the major question this paper intends to answer).

Minor points

The paper, especially the abstract can be shortened to make it more readable.

It is helpful to report R2 in the regression to show how much can be explained by the factors measured in this study.

Typos: by conducting an a priori power analysis

6. PLOS authors have the option to publish the peer review history of their article (what does this mean?). If published, this will include your full peer review and any attached files.

Reviewer #1: No

Reviewer #2: No

---

## [Author Response · Author response to Decision Letter 0]

5 Apr 2022

Please see the enclosed document 'Response to Reviewers' for a properly formatted response.

Dear Editor/Reviewers,

Thank you for the opportunity to revise this manuscript based on the comments outlined in detail below. In this letter, we provide all comments, our response to these comments, as well as an example of how we have implemented this change in our manuscript (if applicable), so that the time to go back and forth between this document and the revised manuscript is minimised. However, note that this only reflects a part of the changes made.

Both reviewers indicated that the research question and the methods were scientifically interesting and that the analyses were conducted well. However, both reviewers outlined some worries regarding interpretation, framing, and missing analyses. In this revised manuscript, we hope to have addressed all reviewer comments in full.

Overall, every section of the previous manuscript has been reworked to significant extent, while parts of the manuscript have been fully rewritten (see Revised Manuscript -Tracked Changes). The main changes that we have implemented are: (i) a fully reworked literature and hypotheses sections that better motivate our research question and the inclusion of all explanatory variables, (ii) additional Bayesian analyses to provide further evidence in favour of a null effect, (iii) a refocusing towards Final Choice as well as Main Choice as equally central tests of our hypothesis, (iv) a requested set of additional regressions, and (v) an additional auxiliary study.

Smaller changes have also been made throughout, focusing on readability and structure of the paper, correct contextualisation within the background literature, further justification of our design choices, as well as additional minor analyses.

We hope that our revisions are satisfactory and that our manuscript can now be considered for publication in PLOS One. However, should there been any further changes and amendment requested, we would gladly take them on board and revise our manuscript further. Thank you very much again for your time and effort in helping us improve this manuscript! 

Reviewer #1

Summary Comment:

In this article, the authors examined the effect of several individual difference measures, among them, risk and ambiguity preferences, numeracy, empathy, optimism, and donation motivation. The main dependent variable is if and if then how much participants were willing to donate from their experimental endowment to one of two different types of charities, which the authors call sure-thing or probabilistic charities. Another dependent variable was the donation to an individual charity (either sure-thing or probabilistic) at a later stage of the experiment. In addition, the authors examined whether an intervention, namely advertising the EV-maximizing principle as a decision criterion, has an effect on donation choices and whether presenting anonymous and context-free donations rather than real-world charities with concrete description of their cause lead to different behavior than the main condition. As results, none of the individual difference measures significantly affected the choice of donating to a sure-thing or probabilistic charity, and only empathy and donation type affected overall donation in the choice to donate to an individual charity in some models.

The paper is clearly written and reports all methods, hypotheses and analyses in an understandable fashion. I appreciate that the study was pre-registered and I think the general research question is important and interesting. Moreover, the individual difference measures are all taken from previous research and seemed to be reasonable choices to measure the respective constructs. However, I am not fully convinced of the classification into sure-thing and probabilistic charity, of the theoretical underpinning of the hypotheses, and the power of the study. I will discuss each of these issues in more detail in the next section.

Comment 1:

The main dependent variable is the choice between so-called sure-thing and probabilistic charities. In total there were six charities, three classified as sure-thing and three classified as probabilistic. This classification is rather ad-hoc and neither rooted in empirical evidence, nor based on principled theoretical arguments. Whereas I might agree from reading the description that this classification could be justified, the current study did not establish that all or even the majority of participants perceived the charities as assumed in the classification. Instead, there is a lot of information in the descriptions of the charities and participants might focus on different information than impact probability. Second, participants could perceive other information about the charity as varying in riskiness. For example, the direct money transfer could be perceived as risky with respect to the ultimate use of the donated money. Finally, the charities in the different conditions could also differ on other dimensions than riskiness. For example, the sure-outcome charities all have a direct impact on individual people, whereas the probabilistic charities have a more indirect impact on a large amount of people. To be fair, the authors mention concerns about the classification of charities into sure-thing and probabilistic in the discussion. However, the provided evidence in favor of their classification are not convincing as they do not directly speak towards a difference in perceived riskiness between the two types of charity. Instead, a straight-forward way to examine this question empirically would be to ask participants either from the old participant pool if available or a new one, how much they rate the respective charities on several dimensions including the riskiness of the project.

Response 1: Thank you very much for raising this critique and for suggesting such an actionable step. As a first step, we have improved on our theoretical justification of this distinction: We have elaborated in much more detail than before for why one might think theoretically that these charities can be divided into these two groups. Furthermore, following your recommendation we have now conducted an auxiliary study (n=101) where participants rated the charities based on how risky their interventions were (riskiness) and how quantifiable they thought these interventions were (ambiguity). We found that probabilistic charities were rated as significantly more risky than sure-thing charities (p<.001) and that they were also rated as significantly more difficult to quantify (i.e. as more ambiguous) (p<.001), indicating that participants do in fact pick up on the underlying dimension of variation in probability and ambiguity in the predicted direction. Taken together, we believe that this distinction is now much better justified than in the original manuscript both theoretically and empirically. This, we argue, allows for a relatively straight interpretation of the results and we hope that this allays most of your concerns about this distinction. However, we will remain to call attention to this potential issue in our discussion section to ensure that all readers are made aware of the potential design weakness and the evidence presented here such that they can form an adequate picture of the data presented in this paper. However, recall also that moving the level of riskiness from easily controllable aspects like donation in standard lab settings to the actual charities themselves is one of the contributions of this paper. 

Example (p. 34-35):

Appendix A – Auxiliary Study

We conducted a post-hoc auxiliary study to empirically confirm our categorisation into sure-thing and probabilistic charities. This is to ensure that this distinction is not only theoretically grounded but also perceived as intended by the general public. We recruited a total of 101 participants on Prolific that had not participated in the main study, none of which failed the attention check. Participants were paid £0.75 for their participation. They were presented with all six charities and were asked to rate them on a scale from 0-10 on the likelihood that the charity’s intervention succeeds (relating to uncertainty over its interventions) and on the quantifiability of the charity’s intervention (relating to ambiguity). We also asked participants to rate the charities on their moral deservingness to keep the objective of this study opaque.

 We find strong support for the distinction between sure-thing charities and probabilistic charities on the basis of both uncertainty and ambiguity. See Appendix Table 1 for means, standard deviations, and medians of the uncertainty and ambiguity ratings for all six charities, with 0 indicating low probability that the charity’s intervention will succeed and a low level of quantifiability of its interventions, and 10 indicating a high probability and quantifiability. In other words, the higher the scores, the less risky and the less ambiguous the charity’s respective intervention is. 

APPENDIX TABLE 1—PROBABILITY-RATINGS FOR ALL SIX CHARITIES

Uncertainty 

 Mean (SD) Median

Sure-Thing Charity 1 (SCI Foundation) 7.19 (2.08) 8

Sure-Thing Charity 2 (GiveDirectly) 5.92 (2.09) 6

Sure-Thing Charity 3 (Against Malaria Foundation)

 7.36 (1.83) 8

Probabilistic Charity 1 (Machine Intelligence Research Institute)

 4.15 (2.46) 4

Probabilistic Charity 2 (Nuclear Threat Initiative) 

fd 4.09 (2.64) 4

Probabilistic Charity 3 (The Center for Health Security) 4.30 (2.50) 4

Ambiguity 

 Mean (SD) Median

Sure-Thing Charity 1 (SCI Foundation) 7.28 (1.97) 8

Sure-Thing Charity 2 (GiveDirectly) 5.42 (2.49) 5

Sure-Thing Charity 3 (Against Malaria Foundation)

 7.24 (2.10) 8

Probabilistic Charity 1 (Machine Intelligence Research Institute)

 3.63 (2.72) 3

Probabilistic Charity 2 (Nuclear Threat Initiative)

fd 4.03 (2.73) 4

Probabilistic Charity 3 (The Center for Health Security) 4.69 (2.40) 4

Notes: Mean, Standard Deviation, and Median of risk and ambiguity ratings for all six charities.

We find that the data behave as generally expected, with sure-thing charities receiving higher ratings about the likelihood that their interventions will succeed as well as higher ratings for the quantifiability of their interventions, and probabilistic charities receiving lower rating correspondingly. Adding a subject’s scores for the individual charities of each bucket, we find that sure-thing charities are rated as having significantly higher probability interventions (M=20.47, SD=4.52) than the probabilistic charities (M=13.96, SD=5.83). This difference, 6.51, 95% CI [5.19, 7.82] was highly statistically significant, t(100)=9.81, p<.001. The same picture emerges with regard to the quantifiability of the interventions, with the mean of the sum of the quantifiability scores of sure-thing charities (M=19.83, SD=4.60) being significantly higher than that of the probabilistic charities (M=12.85, SD=6.70), with the difference of 6.98, 95% CI [5.62, 8.34] also being statistically significant at t(100)=10.17, p<.001. The effect sizes of these two differences, in Cohen’s d, is d=.98 for the probability ratings and d=1.01 for the quantifiability rankings. This provides strong support for our theoretically based distinction between sure-thing and probabilistic charities. 

Comment 2:

Since there were different charities administered in the stimulus material, it would make sense to estimate regressions with stimulus random effects, instead of OLS. Other models that cluster errors on the stimulus level might also be possible. Similar to the logistic, this could be done as a robustness test.

Response 2: Thank you very much for suggesting this analysis. We agree that this type of regression would be appropriate for our data structure. We now report one such model in the appendix (Appendix Table 3, Model 5), where we find the same pattern of results as in the main analyses, suggesting that our results are also robust to this modelling choice. 

Example (p. 36-37):

We also report an additional robustness check of Model (1) in Appendix Table 3, Model (5). Specifically, we report a random effects model with the stimulus material being treated as a random effect. As we had three sure-thing and three probabilistic charities, there were nine charity pairs that participants could have been presented with. In Model (5), we treated the stimulus (i.e. the charity pair presented) as a random effect (with nine levels). Because significance levels are not unproblematic in mixed models like this, we also report 95% confidence intervals of the estimates. The results indicate that our null effect is robust to this model choice as well. 

APPENDIX TABLE 3— REGRESSION RESULTS FOR MAIN CHOICE – RANDOM EFFECTS ROBUSTNESS CHECK

PREDICTING CHOICE BETWEEN SURE-THING AND PROBABILISTIC CHARITIES 

 (5) 

Risk Attitude .002 (.005) [-.008, .012]

Ambiguity Aversion -.002 (.005) [-.010, .006)

Numeracy .003 (.023) [-.042, 0.47]

Empathy .001 (.002) [-.005, .004]

Optimism

 -.006* (.003) [-.012, .001]

Donor Type 

 Warm-Glow -.007 (.047) [-.099, .086]

 Pure Altruism -.040 (.059) [-.156, .077]

Age .001 (.002) [-.003, .004]

Gender -.024 (.047) [-.115, .068]

Education 

 Undergraduate degree .041 (.048) [-.053, .134]

 Postgraduate/Professional degree -.023 (.051) [-.123, .078]

Religion 

 Protestantism -.043 (.062) [-.165, .080]

 Catholicism .020 (.068) [-.114, .154]

 Islam -.035 (.113) [-.246, .188]

 Judaism -.006 (.354) [-.691, .703]

 Buddhism -.060 (.265) [-.581, .462]

 Hinduism .325 (.254) [-.175, .825]

Religious Participation -.004 (.083) [-.168, .159]

Marriage Status -.019 (.046) [-.071, .108]

Children .009 (.050) [-.090, .107]

Financial Wellbeing .009 (.022) [-.035, .053]

Employment 

 Part-time employment .058 (.054) [-.047, .164]

 Full-time employment .015 (.049) [-.083, .112] 

Sample size 307 

Notes: Random effects model, coefficients, and standard errors, as well as 95% CIs. *p<.1, **p<.05, ***p<.01, ****p<.001

Comment 3:

More effort should be spent on the theory of why the examined individual differences should be correlated with the choice between the sure-thing and the probabilistic charity. In particular, why should risk preference be related to this choice, if—as mentioned by the authors—previous literature (Vives & FeldmanHall, 2018) did not find a connection between risk-preference and prosocial behavior, that the exact probabilities of success of a charity are most likely ambiguous and that risk- and ambiguity preferences are not strongly correlated? Similarly, the connection between optimism or numeracy and the main DV is not build on theoretical considerations nor on previous findings that suggest such a connection. A strong foundation of the hypotheses would considerably strengthen the importance of the article.

Response 3: Thank you very much for pressing us on this. We agree that our explanations were not thorough enough in the previous manuscript. We have now expanded upon our justification for all our individual difference measures throughout the revised manuscript, making clearer our reasoning for their inclusion. Specifically, we have now backed up the inclusion of all items with a wider number of references to the literature, showing how all constructs have been previously found to be associated with some measure of pro-social behaviour or charitable giving to directly motivate and justify their inclusion. Further, we have also updated our analytical reasoning to further improve the justifications of inclusion on both counts.

Example (p. 6-7):

First, we investigate whether donor choices can be explained by individual differences in risk and ambiguity attitudes. Previous work in domains such as stock market participation (Bianchi & Tallon, 2018) and health-related field behaviours (Sutter et al., 2013) has found that attitudes to risk and ambiguity can play significant roles. In context of pro-social behaviour in game environments the results show that ambiguity aversion may play a role (while risk aversion sometimes does not) (Vives & FeldmanHall, 2018), though risk aversion has also been found to be “predictive for giving” (Cettolin, Riedl, & Tran, 2017, 95). As such, we argue that given risk and ambiguity aversion have been shown to impact behaviour in many contexts including charitable giving, this makes it an a priori interesting relation to test. This hypothesis is also theoretically grounded, in that it might be the case that one’s preference not to give to charities that have a low chance of making an impact might be driven by an individual’s general risk aversion profile, or it might be that given the ambiguous nature of charitable interventions that it is only ambiguity aversion that impacts this choice. 

[…]

For example, previous research has found that those lower in numeracy were more insensitive to proportions of donation recipients (Kleber, Dickert, Peters, & Florack, 2013) and that they showed higher susceptibility to changes in numeric presentation (Dickert, Kleber, Peters, & Slovic, 2011). It may as such be the case that one’s level of numeracy also meaningfully impacts behaviour in the context studied here as the probabilistic charities include interventions that have a small chance of making a large impact. Understanding these proportions plausibly requires a certain level of numeracy. Further, one may also think that a general proclivity to optimism may bias individuals towards overestimating the success of probabilistic charitable interventions, or conversely that higher pessimism may explain a preference for sure-thing charities as those promise to have a reliable impact even in the worst-case scenario. This is corroborated by previous research that draws on the German socioeconomic panel and finds that optimism predicts charitable giving in some of their models (Boenigk & Mayr, 2016).

Comment 4:

There is some uncertainty about the power of the regression results as the pre-registered power analyses were based on the assumption that 2/3 donated, but the true number of donations was only 1/3. I appreciate that this is mentioned in the discussion, but I think this aspect deserves a bit more attention. In particular the EV-max intervention and the context-free manipulations with a much smaller sample size than the main dependent variable, might not have enough power to draw definite conclusions. Instead, it might make sense to put these analyses in an appendix and label these analyses as inconclusive. 

Response 4: Thank you for your comment and the actionable advice. We agree with you that the context-free manipulations (and in some respects the EV-information interventions) might have a too-low sample size to draw conclusions from justifiable. We have now followed your suggestion and labelled these as ‘inconclusive’ and have put them in the appendix. However, we do not share the same pessimism about Main Choice: Recall that our original power analysis was extremely strict, having been aimed to have .95 power to detect at f2=.02. While we understand the statistical challenges of calculating power post-hoc, moving from .95 power to .8 in the above power analysis would have reduced the sample size needed to nearly a fifth, making our collected sample size much more reasonable with this in mind. Also, our data from Final Choice does not share these concerns, and following Reviewer 2, we have now increased the prominence of our discussion of the results from Final Choice. 

Example (p. 17):

However, because the number of people who made donations was unexpectedly small, both of these conditions did not have the power that we calculated prior to running this study to detect a meaningful effect. This means that results of these conditions are inconclusive. We still report the full pre-registered analyses in the appendix, see Appendix D, but do not discuss them in the main results and discussion sections.

Comment 5: For the main DV, I appreciate that the authors calculate equivalent tests to examine the measurement uncertainty around the true effect. In my view, a better or at least complementary approach would be to calculate Bayes factors to evaluate the evidence for the Null (see Jarosz & Wiley, 2014; Rouder & Morey, 2012). Bayes Factors could clearly state whether enough evidence has been collected to conclude that there is no effect of an individual difference measure on the choice between the sure-thing or the probabilistic charity. In case that the evidence is inconclusive, it might make sense to collect more data.

Response 5: We are very glad that you appreciate our use of equivalence tests. We have now expanded our usage of equivalence tests (now also for the results from Final Choice) to provide the reader with a wide range of plausible test ranges that may allow us to conclude a null effect (and to narrowly estimate the range of the equivalence bounds). Further, we have also followed your suggestion and included Bayesian analyses following Rouder & Morey (2012). The Bayesian results corroborate our equivalence test results in that they provide strong evidence in favour of a null effect. 

Example (p. 24):

Following Rouder & Morey (2012), we compute exploratory Bayes factors to evaluate the evidence for the null. We report Bayes factors for linear models based on Liang et al. (2008), using a Jeffrey-Zeelner-Siow mixture of g-priors. Applying the default r scale of .353 for the results regarding Main Choice in Model (1), we calculate a JZS Bayes Factor of 2.49*107, suggesting very strong evidence in favour of the null hypothesis. The same analysis regarding Final Choice in Model (2) also provides evidence in favour of the null, though at a significantly smaller magnitude, at a JZS Bayes Factor of 22.15. This corroborates the results of the equivalence tests and suggests that our data provide strong evidence in favour of a robust null result. 

Comment 6:

I would suggest to describe the hypotheses in the introduction in terms of the alternative hypothesis. Presenting hypotheses as the Null makes the text wordier and more complex than necessary.

Response 6: Thank you for your comment! While we agree that statements in terms of alternative hypotheses would aid comprehension in some respects, we will retain the phrasing in terms of null hypotheses for the following reasons:

1) We have pre-registered the null hypotheses this way and would like to stick as closely as possible to our pre-registration.

2) More importantly, because the results in this paper are primarily null results, retaining the null hypothesis structure enables us to more meaningfully discuss our inability to reject the null and to discuss evidence in favour of the null (with equivalence tests). As such, while alternative hypotheses might be easier to understand in the earlier sections of the paper, we believe that when it comes to the statistical results and discussions of them, null hypothesis phrasings are actually more straightforward. 

Comment 7:

On page 5 it is stated that:

“Our research builds on this literature but is importantly different, primarily because of our focus on donations to actual charities and not on pro-social behaviour in abstract games.”

And on page 6:

“Specifically, we focus on donation behaviour between real world charities that are made with an earned endowment, in contrast to abstract laboratory game pro-social decisions and hypothetical choices.”

These comments suggest that there is basically no research about charitable giving and all research about social preferences is only done in abstract experimental paradigms. I think this is a wrong impression and literature about charitable giving should be properly cited in these situations. Possible literature and review articles of which some are also cited in the text are Bekkers and Wiepking (2011), Karlan and List (2007), and Vesterlund and Sonnevi (2007).

References

Bekkers, R., & Wiepking, P. (2011). A literature review of empirical studies of philanthropy: Eight mechanisms that drive charitable giving. Nonprofit and voluntary sector quarterly, 40(5), 924-973.

Jarosz, A. F., & Wiley, J. (2014). What are the odds? A practical guide to computing and reporting Bayes factors. The Journal of Problem Solving, 7(1), 2.

Karlan, D., & List, J. A. (2007). Does price matter in charitable giving? Evidence from a large-scale natural field experiment. American Economic Review, 97(5), 1774-1793.

Rouder, J. N., & Morey, R. D. (2012). Default Bayes factors for model selection in regression. Multivariate Behavioral Research, 47(6), 877-903.

Vesterlund, L., & Sonnevi, G. (2006). 24. Why Do People Give?. In The nonprofit sector (pp. 568-588). Yale University Press.

Vives, M. L., & FeldmanHall, O. (2018). Tolerance to ambiguous uncertainty predicts prosocial behavior. Nature communications, 9(1), 1-9.

Response 7: Thank you very much for pressing us on this! We of course did not mean to suggest that there was no research fitting these criteria. We have now adjusted our language throughout the literature review and cited all the literature outlined by you (except the ones that we had already cited before), also clearing up some additional issues in these sections: Specifically, we now make explicit that our contribution is primarily moving risk over donations in tightly controlled abstract games (as has been done before) to the risk/ambiguity over actual charities’ interventions, something that has both not been studied in detail and is much closer to the actual choice environment that potential donors find themselves in. Now we make clear that our claim to increase ecological validity is with regards to this aspect specifically, and of course not to charitable giving overall (though we agree that our previous manuscript did not make this sufficiently clear). We hope that our updated language now removes all ambiguity regarding this issue.

Example (p. 4-5):

Our research builds on the literature on risk that has so far mostly employed directly controllable levels of risk in the lab. For example, in abstract game scenarios, risk can be controlled and stated precisely, for example by imposing a 50% chance of one’s donation not going through, or by introducing a 5% chance that one’s donation is matched. Crucially though, our research is substantially different from the discussed literature primarily because we move the level of risk from directly calculable interventions in the lab (as outlined above) to the actual charities themselves. While this introduces several design challenges, we argue that this step leads to an increased level of ecological validity of any potential finding. However, note that there is already a large literature on charitable giving generally that has a similar or higher level of external validity (Bekkers & Wiepking, 2011; Karlan & List, 2007; Vesterlund & Sonevi, 2006). However, our paper’s main contribution is the moving of our focus on risk and ambiguity to actual organisations and their interventions and away from aspects that can be controlled in the lab. Having risk and ambiguity at the level of actual charities is the level at which risk and ambiguity typically enter people’s decision-making processes; rarely are we uncertain as to whether our donation will randomly increase when we donate (as in some experimental lab studies), but we are almost always acting under uncertainty about the charity’s interventions that we consider donating to. 

Reviewer #2:

Summary Comment:

While I was reading the paper, I found the research questions interesting and important. I liked it a lot. However, when I read though the experiment design, I found, unfortunately, the design in the Main choice and context-free choice is not appropriate to answer the research questions. The concern is that the current design asks the participants to make two choices: which type of charity to donate to and how much to donate, while these two decisions are correlated with each other. Consequently, the experiment is not well controlled.

Comment 1:

To make this more clear, I’ll first describe what is an ideal design and then point out the issues about the current design. To answer the research questions or test the hypotheses, we can either test whether they prefer to donate to sure-thing charity or probabilistic charity or test how much they would like to donate given sure-thing charity or probabilistic charity. In the former case, we need to control for the donation level, i.e. if they donate, they donate the same amount of money. In the latter case, we randomly control for the charity type. The latter case is of course the Final Choice design in the current paper which I fully agree is the correct way.

The issue with the current design is that, theoretically, there should be a point where one is indifferent between donate a certain amount of money under sure-thing charity and another amount under probabilistic charity. So what we watched in current experiment is just one of the two choices they are indifferent from. Therefore, it is hard to tell from their preference for the charity type.

To extend the above point a bit more and perhaps it helps to make it more clear, suppose there are some amount of people who would donate zero no matter what type of charity they are assigned or chose in the Main choice (this happens I guess for sure because there are only 35.8% of participants made a donation). For these people, economically, their choice of the charity type is invalid as the cost is zero (they will not donate anyway). This is an extreme case.

Now return to the paper, to save it, first, I think the Final Choice design is good, the authors may want to rely more on the data generate from the Final Choice. But at the same time take it in mind that this is last task which may suffer from order or experimenter demand effect. 

Response 1: Thank you very much for your thoughtful comments. We actually agree with you that the design for Final Choice is superior to that of Main Choice, primarily because Main Choice is a more substantive (and thus more confoundable) design, compared to the overall much cleaner design of Final Choice. As such, we have followed your recommendation and have prioritised the data from Final Choice. Specifically, we now report Main Choice and Final Choice results next to each other throughout the entire manuscript and discuss their comparative strengths and weaknesses in the discussion section. Further, on suggestion of Reviewer 1, we have also moved the results and discussion of the no-context condition as well as the expected-value treatment to the appendix, giving the Final Choice data even more space (effectively doubling its share of discussion space in the main manuscript). However, we will not solely rely on the data from Final Choice as we pre-registered our main hypotheses with both Main Choice and Final Choice in mind, and because we do believe that the data from Main Choice do meaningfully contribute to our understanding of the research question. We hope that our restructuring and comparative focus on Final Choice satisfactorily responds to your actionable request to rely more on Final Choice. 

Lastly, we believe that given we have understood your worry correctly, we have some empirical evidence from our data that should go towards alleviating your worry to some extent. We believe that one testable prediction from your worry would be that those who donated to sure-thing charities before (i.e., in Main Choice) should donate less to probabilistic charities at Final Choice. We have investigated this in our data, looking at participants who donated to a sure-thing charity in Main Choice and correlated these donations with their donation amounts (including 0) when they were presented with a probabilistic charity. We do not find a statistically significant correlation at r(184)=.087, p=.238. We also do not find the predicted negative correlation in the reverse (with participants who donated to a probabilistic charity first and were then shown a sure-thing charity in Final Choice, though these results rest on a very low sample size and should probably be disregarded), with r(25)=.442, p=.027. These results suggest that your worry may not impact the data as much as one might have thought, making our solution of presenting both Main Choice and Final Choice as central pieces of the paper justified. Further, when you claim that one piece of your worry is that some people would “donate zero no matter what type of charity they are assigned or chose in the Main choice” we just want to quickly point out that those people would not be included in any of the analyses of Main Choice as only those making a donation are included in the regression models based on our pre-registered exclusion criteria.

Example (p. 6; 24-25):

In our experiment, each participant is first presented with a randomly selected pair of charities consisting of one charity of each type to control for accidental confounds relating the charity’s context as each are presented with substantial additional accurate information to increase the naturalness of the choice. Participant choices with respect to this randomly selected charity pair then allows us to isolate and capture the element of probability between the two charity types. In the second part of this experiment, we study participant behaviour when they are shown only one randomly selected charity (either sure-thing or probabilistic), which more narrowly captures the predictive value of individual differences on donation choices to charities of specific types. Overall, we find little to no evidence that individual differences in risk/ambiguity attitudes, numeracy, optimism, and donor type predict charitable giving behaviour. However, we do find that a purely altruistic donor type predicts donations to probabilistic charities. As such, we take this paper to be primarily reporting a null-result. 

[…]

Overall, we find little or no evidence in favour of rejecting our null hypotheses, and as such this paper primarily reports a negative result: First, we do not find evidence that risk and ambiguity attitudes robustly predict charitable decision-making between sure-thing and probabilistic charities (null hypothesis #1), and we also do not find evidence that individual differences in numeracy, optimism, donor type, and empathy predict behaviour in this choice as well (null hypothesis #2). Further, we only find weak evidence predicting behaviour when participants are presented with either a sure-thing or a probabilistic charity, where we find that pure altruistic donor types significantly predict donation frequency to probabilistic charities (null hypothesis #5). Further, due to a lower than planned sample size, we are unable to conclusively evaluate the data about our no-context condition (null hypothesis #3) and with regard to the effect of the informational treatment introduced participants to expected-value reasoning (null hypothesis #4). 

[…]

As such, it may be justified to put more interpretative emphasis on the results from Final Choice compared to Main Choice.

Comment 2:

Second, if the authors really want to use the data from the Main Choice, they should at least control for the donate amount in the regression. Though I still think this is not valid enough to test the hypotheses.

Response 2: Thank you for your comment. However, we have decided not to follow it because we are mindful of introducing endogeneity in the regression models by having two dimensions of the same choice in the same model as independent and dependent variables (choice of charity as Dep. V and size of that donation choice as Ind. V). If there is an opportunity for further revision, could you please elaborate for why you think we should include this control variable despite the challenges of endogeneity (and how we could circumvent these)? Then we’d be more than happy to follow your advice.

Comment 3:

In table 4, to test the hypothesis using the Final Choice data, they should add in interactive term between treatment variable (sure-thing or probabilistic) and the major explanatory variables to test the difference in difference, that is whether the major explanatory variables can explain the donation difference under sure-thing or probabilistic charity (the major question this paper intends to answer).

Response 3: Thank you very much for this comment! We agree that this would be a better approach and have moved our original (pre-registered) regressions to the appendix and now report two regressions with the interaction terms as proposed by you in the main text and rely on their results throughout the paper. We also use the interaction terms for our equivalence tests of the Final Choice coefficients.

Example (p. 21-22):

Second, we investigate general donation behaviour in Final Choice where participants were either presented with a sure-thing or a probabilistic charity. Here, we report two further regression models relevant to null hypothesis #5. These are not the pre-registered ones but instead include interaction terms that we did not pre-register. For specifications of the regression models as pre-registered, see Appendix C (Appendix Table 6). Model (2) explains frequency of donation, and Model (3) predicts size of donation. For Model (2), not making a donation is coded as 0 and making a donation is coded as 1. The central variables are interaction terms, where we interact the main individual difference measures with the condition. Specifically, they are coded with 0 = sure-thing charity and 1 = probabilistic charity for all our main explanatory variables. In these analyses, participants from all conditions are included and are split only by which type of charity they were presented with at Final Choice; recall that each participant here was only presented with one randomly selected charity. 

TABLE 3—REGRESSION RESULTS FOR CHARITABLE GIVING BEHAVIOR IN FINAL CHOICE

PREDICTING FREQUENCY OF GIVING AND SIZE OF DONATION

 (2) (3)

Risk Attitude .002 (.005) .124 (.120)

Ambiguity Aversion -.002 (.004) -.127 (.109)

Numeracy .015 (.017) -.193 (.453)

Empathy .007**** (.002) .212**** (.048)

Optimism

 .003 (.003) .062 (.071)

Donor Type 

 Warm-Glow .061* (.047) 2.170* (1.237)

 Pure Altruism .045 (.058) -.144 (1.525)

Condition X Risk Attitude .003 (.006) .019 (.161)

Condition X Ambiguity Aversion .002 (.005) .085 (.143)

Condition X Numeracy .006 (.018) .275 (.482)

Condition X Empathy -.002* (.002) -.086 (.056)

Condition X Optimism -.004 (.003) -.033 (.088)

Donor Type 

 Condition X Warm-Glow -.034 (.063) -.993 (1.649)

 Condition X Pure Altruism .177** (.083) 2.955 (2.176)

Age .001 (.001) .047 (.030)

Gender -.050* (.030) -.747 (.788)

Education 

 Undergraduate degree -.050 (.031) -1.129 (.800)

 Postgraduate/Professional degree .030 (.034) -.095 (.897)

Religion 

 Protestantism .071* (.043) .153 (1.129)

 Catholicism .045 (.047) 1.468 (1.239)

 Islam .173* (.142) 6.606*** (2.317)

 Judaism .134 (.142) 2.636 (3.730)

 Buddhism .159 (.160) 4.994 (4.207)

 Hinduism .149 (.126) 7.049** (3.301)

 Sikhism .046 (.260) -2.343 (6.819)

Religious Participation .015 (.054) -.265 (1.406)

Marriage Status -.025 (.031) -.469 (.819)

Children .032 (.033) .697 (.877)

Financial Wellbeing .023 (.015) .398 (.384)

Employment 

 Out of the workforce .017 (.055) -.820 (1.442)

 Part-time employment -.019 (.047) -2.211* (1.243)

 Full-time employment -.029 (.042) -1.436 (1.107)

R2 .076 .085

Sample size 1177 1177

Notes: OLS regressions reporting unstandardised coefficients and standard errors. Model (2) predicts frequency of donation and Model (3) predicts size of donation. Interaction terms interact the condition (0 = sure-thing charity, 1 = probabilistic charity) with the explanatory variables *p<.1, **p<.05, ***p<.01, ****p<.001

Here we find that while empathy predicts donation frequency and size overall, it is only the interaction term with pure altruism that statistically significantly predicts frequency of donation to probabilistic charities. This is in line with theoretical predictions that hold that pure altruists would be more likely to give to probabilistic charities as they primarily care about the impact on social welfare. Further, in our robustness check that reproduces Model (2) as a logit model (Appendix C, Appendix Table 5), we find the same result. All other pre-registered variables (interacting with condition) do not predict donor behaviour. 

Comment 4:

The paper, especially the abstract can be shortened to make it more readable.

Response 4: Thank you for your comment. In this revision we have tried to make the paper shorter throughout (and focused especially on the abstract). Most of the robustness checks that were requested by the reviewers are now in the appendix to ensure that the paper’s length is not massively increased by responding to reviewer comments. The paper without references and appendix is now under 30 pages. 

Comment 5:

It is helpful to report R2 in the regression to show how much can be explained by the factors measured in this study.

Response 5: Thank you very much for suggesting this. We now report R2 (or Cox and Snell R2) for all our regressions.

Comment 6:

Typos: by conducting an a priori power analysis

Response 6: Apologies, but we are unsure what the typo here is.

---

## [Decision Letter · Decision Letter 1]

16 May 2022

PONE-D-21-35635R1Sure-Thing vs. Probabilistic Charitable Giving: Experimental Evidence On the Role of Individual Differences in Risky and Ambiguous Charitable Decision-MakingPLOS ONE

Dear Dr. Schoenegger,

Thank you for submitting your manuscript to PLOS ONE. After careful consideration, we feel that it has merit but does not fully meet PLOS ONE’s publication criteria as it currently stands. Therefore, we invite you to submit a revised version of the manuscript that addresses the points raised during the review process.

We look forward to receiving your revised manuscript.

Kind regards,

Junhuan Zhang, PhD

Academic Editor

PLOS ONE

Journal Requirements:

Reviewers' comments:

Reviewer's Responses to Questions

**Comments to the Author**

1. If the authors have adequately addressed your comments raised in a previous round of review and you feel that this manuscript is now acceptable for publication, you may indicate that here to bypass the “Comments to the Author” section, enter your conflict of interest statement in the “Confidential to Editor” section, and submit your "Accept" recommendation.

Reviewer #1: All comments have been addressed

Reviewer #2: (No Response)

2. Is the manuscript technically sound, and do the data support the conclusions?

Reviewer #1: Yes

Reviewer #2: No

3. Has the statistical analysis been performed appropriately and rigorously? 

Reviewer #1: Yes

Reviewer #2: No

4. Have the authors made all data underlying the findings in their manuscript fully available?

Reviewer #1: Yes

Reviewer #2: Yes

5. Is the manuscript presented in an intelligible fashion and written in standard English?

Reviewer #1: Yes

Reviewer #2: Yes

6. Review Comments to the Author

Reviewer #1: In general, the authors incorporated all my comments and improved the paper considerably. I would like to thank and congratulate them for their work and their constructive responses to the raised comments. I think the paper can be published with only minor revisions:

1. I very much like the additional study and the demonstration that participants indeed perceived the one class of charities as riskier and more ambiguous than the other. Given the importance of this result in judging the whole validity of the study with respect to the research question, I feel this study deserves much more space in the main manuscript. I think it should get a small methods and results section (maybe called study to validate the stimuli) and should also be mentioned already in the Methods of the main study when the different charities are introduced as stimuli. Importantly, while the additional study helps to support the claim that the charities differ in perceived riskiness, it does not exclude the possibility that the groups of charities also differed on other dimensions (e.g., individual vs. group as recipients etc.). I think this is an important limitation that should be mentioned in the discussion.

2. It is great that the authors conducted a Bayes Factor analysis. However, it was not exactly clear to me which models the authors compared. If it were the full model with all predictors vs. the null model with only an intercept this should be clearly reported and interpreted accordingly. Given the hypotheses, it would make sense to compare all individual models with one predictor from the five individual difference measures each against the null model with just an intercept. That way, the evidence for the individual null hypotheses could be unambiguously assessed.

3. Again, I think the authors did a good job in adding more literature and theoretical arguments to justify why the selected measures should correlate with donation behavior in their study. However, I also noticed that most of the studies they cite are based on different experimental designs. For example, Kleber et al. (2013) found an effect of numeracy on the effect of the number and share of helped people. This feature is not central to the current manipulation of interest. Similarly, in Cettolin et al. (2017) risk preferences are measures with certainty equivalents in decisions from description, whereas the current study uses an experience-based risk elicitation task with a different dependent variable. We already know that behavioral elicitation methods for risk preference correlate little with each other (see Frey et al., 2017). Thus, it is not straight-forward to assume the same underlying relation from different elicitation methods. Ultimately, I think this is not a weakness of the current study, but rather of the field as a whole. I would suggest the authors mention this problem in the discussion and call for more specific cognitive models and theories that build stronger connections between elicitation methods, constructs, and cognitive processes influencing behavior.

References:

Frey, R., Pedroni, A., Mata, R., Rieskamp, J., & Hertwig, R. (2017). Risk preference shares the psychometric structure of major psychological traits. Science Advances, 3(10), e1701381.

Reviewer #2: I appreciate the authors effort to give more weight to Final Choice. The paper has been improved a lot for sure. However, I’m not sure whether the authors understand my major concern well. The point is that you have two moving parts in your Main Choice design. Such a design flaw makes it unable to test your hypotheses and hence is totally invalid. The reasoning is simple: when one can choose both the type of the charity and the amount to donate, one can choose two different compositions that one is indifferent from. For example, if I’m indifferent between $1/sure thing charity and $0.5/probabilistic charity, I can choose randomly from the two. Overall, you would find roughly 50% choose sure thing and 50% choose probabilistic charity. This means you cannot test the preference for charity type under such a design. This is point one. Point two, suppose people are more likely to donate more under sure-thing charity, then when you excluded 0 donation as you preregistered, you exclude more people who chose probabilistic charity (you can test this with your data). Th reason is that there must be some people who are indifferent between $positive amount/sure thing charity and $0/probabilistic charity, and these people chose randomly between the two. In the observed results, you excluded those who chose probabilistic charity but not the other type. Point 3, because of the above issue, if you regress charity type preference on other variables, you have to control for the amount donated, as an ex-post control, because you did not control for the donate amount in the experiment. I’m not clear what endogeneity this may cause. But if there is any, I don’t believe the data from Main Choice can prove anything. There is no problem if you want to report the analyses as you preregistered. But a preregistration doesn’t mean your design and analysis (Main Choice) is not problematic.

7. PLOS authors have the option to publish the peer review history of their article (what does this mean?). If published, this will include your full peer review and any attached files.

Reviewer #1: **Yes: **Sebastian Olschewski

Reviewer #2: No

---

## [Author Response · Author response to Decision Letter 1]

23 Jun 2022

[PLEASE SEE ATTACHED FILE FOR A PROPERLY FORMATTED RESPONSE]

Dear Reviewers,

Thank you very much for evaluating our revised manuscript and for giving us the opportunity to revise and improve it further.

We are very happy that both of you agreed that our previous revision “improved the paper considerably” (R1) and “improved the paper a lot” (R2). However, we acknowledge that we have not yet fully addressed all of the comments raised to a satisfactory extent and we hope that in this revision, we will have done so. 

Overall, we have made the following general improvements. 

Following Reviewer 1’s recommendations, we have now moved our auxiliary study into the main text to give it more spotlight compared to the previous appendix placement. We have also updated our Bayesian analyses, now reporting comparisons of all individual models with the intercept-only model. Further, in part in response to Reviewer 2’s comment from the first round of revision, we have also provided model-averaged coefficients for the full Bayesian models of Final Choice to provide further analyses of the condition that suffers less design worries.

Following Reviewer 2’s recommendations, we now control for the amount donated in all analyses of Main Choice (and have updated our equivalence tests accordingly) such that Main Choice’s design downsides are at least in part addressed. In the discussion section, we now also discuss in much detail the major limitation that Reviewer 2 has identified with Main Choice. While we cannot drop these analyses from our paper as they were our primary pre-registered analyses, we now give significant space to this objection such that readers are fully informed about this design choice and its flaws. Further, we now report a regression in the appendix that mirrors Main Choice but aims to control for some of those worries.

Additionally, we have slightly reworked the literature section, updated the parts of the discussion section, and made further minor changes throughout the document.

We hope that these changes are sufficient for this paper to be considered for publication at PLOS One. However, if you have any further improvements that you’d like to see us make, we’d be more than happy to accommodate them too!

Reviewer #1

General Comment: In general, the authors incorporated all my comments and improved the paper considerably. I would like to thank and congratulate them for their work and their constructive responses to the raised comments. I think the paper can be published with only minor revisions:

General Response: Thank you very much! We hope that the changes below properly respond to your requested revisions.

Comment 1: I very much like the additional study and the demonstration that participants indeed perceived the one class of charities as riskier and more ambiguous than the other. Given the importance of this result in judging the whole validity of the study with respect to the research question, I feel this study deserves much more space in the main manuscript. I think it should get a small methods and results section (maybe called study to validate the stimuli) and should also be mentioned already in the Methods of the main study when the different charities are introduced as stimuli. Importantly, while the additional study helps to support the claim that the charities differ in perceived riskiness, it does not exclude the possibility that the groups of charities also differed on other dimensions (e.g., individual vs. group as recipients etc.). I think this is an important limitation that should be mentioned in the discussion.

Response 1: Glad to hear the addition of the auxiliary study has adequately responded to your previous request for revisions. We have now moved the study to the main text as you suggested. We have also mentioned the valid concern you have raised in the discussion section.

Example (p. 30): However, it is worth pointing out that this auxiliary study cannot help us rule out whether the two groups of charities also differed along other dimensions in the perceptions of participants. 

Comment 2: It is great that the authors conducted a Bayes Factor analysis. However, it was not exactly clear to me which models the authors compared. If it were the full model with all predictors vs. the null model with only an intercept this should be clearly reported and interpreted accordingly. Given the hypotheses, it would make sense to compare all individual models with one predictor from the five individual difference measures each against the null model with just an intercept. That way, the evidence for the individual null hypotheses could be unambiguously assessed.

Response 2: Thank you for pointing out this shortcoming and giving a clear suggestion on how to improve it. We have now run a set of Bayesian linear regressions where we compare the individual models (with only one variable of interest) to the intercept-only model. For Final Choice, we do this for the interaction terms. In addition, in part in response to Reviewer 2’s suggestion to focus more on Final Choice, we also report model-averaged coefficients for Bayesian linear regression models with all variables (including controls) entered for the variables of interest, corroborating previous results.

Example (p. 26-27): Further, we conduct exploratory Bayesian analyses (Rouder & Morey 2012; Liang et al. 2008) where we use Bayesian linear regression analyses that draw on Bayesian model averaging (e.g., Hinne et al., 2019, Maier et al., 2022). First, in Table 7, we report results where we compare the individual models with one predictor each (the respective explanatory variable) against the null model with an intercept only. We report Bayes Factor Model Odds for the null for both Main Choice and Final Choice (frequency of donation as well as size of donation). For the analyses in Table 7, we use a uniform model prior.

Table 7—Bayes Factor Model Odds for Null Model

 Main Choice Final Choice (Freq.) Final Choice (Size)

Risk Attitude 7.46 

Ambiguity Aversion 7.81 

Numeracy 7.44 

Empathy 7.89 

Optimism 3.50 

Warm-Glow Type 7.94 

Pure Altruistic Type 5.94 

Condition X Risk Attitude .04 <.001

Condition X Ambiguity Aversion 13.41 7.57

Condition X Numeracy <.001 <.001

Condition X Empathy <.001 <.001

Condition X Optimism <.001 <.001

Condition X Warm Glow 3.82 6.23

Condition X Pure Altruism 1.46 15.32

Notes: Bayes Factor Model Odds for the Null Model with Uniform Prior for both the explanatory variables for Main Choice and the interaction terms for Final Choice.

The results in Table 7 indicate that for Main Choice, we have strong additional evidence in favour of a null effect of all variables of interest with Bayes factors of between 3.5 and 7.94. The picture is more complicated with regard to the interaction effects from Final Choice. There, we find that compared to models with ambiguity aversion or altruistic types individually, the null model is much more likely given the data. However, the other explanatory variables do not share this pattern. To further investigate the effect of our main variables on our outcomes of interest, we also report the model averaged coefficients that account for uncertainty over the estimates as well as uncertainty over model choice with all interaction effects and all control variables (age, gender, education, marital status, children, financial well-being, employment, religious affiliation, and religious participation). We report those coefficients as well as their 95% credible intervals that represent a weighted average that is weighted by the posterior probability of predictor inclusion. We use a uniform model prior and a JZS parameter prior with the default r scale of .354.

Table 8—Bayesian Linear Regression Coefficients and 95% Credible Intervals

 Final Choices (Freq.) Final Choice (Size)

Condition X Risk Attitude .000 [-.001, .005] .02 [-.03, .24]

Condition X Ambiguity Aversion .000 [-.001, .005] -.003 [-.008, .10]

Condition X Numeracy -.01 [-.02, .000] -.49 [-.89, .000]

Condition X Empathy .000 [-.002, .001] .001 [-.07, .06]

Condition X Optimism -.001 [-.005, .000] -.000 [-.03, .08]

Condition X Warm Glow -.004 [-.01, .07] .19 [-.25, 2.19]

Condition X Pure Altruism .20 [.08, .32] .90 [.000, 4.66]

Notes: Model averaged coefficients and 95% credible intervals. 

The results reported in Table 8 are in line with previous frequentist regression analyses, where we find that pure altruistic donor type predicts donations to probabilistic charities while most other interaction effects do not show any effects or only provide weak evidence in favour of them once all variables are entered into the model. As such, we conclude that the exploratory Bayesian analyses roughly corroborate our previous results. 

Comment 3: Again, I think the authors did a good job in adding more literature and theoretical arguments to justify why the selected measures should correlate with donation behavior in their study. However, I also noticed that most of the studies they cite are based on different experimental designs. For example, Kleber et al. (2013) found an effect of numeracy on the effect of the number and share of helped people. This feature is not central to the current manipulation of interest. Similarly, in Cettolin et al. (2017) risk preferences are measures with certainty equivalents in decisions from description, whereas the current study uses an experience-based risk elicitation task with a different dependent variable. We already know that behavioral elicitation methods for risk preference correlate little with each other (see Frey et al., 2017). Thus, it is not straight-forward to assume the same underlying relation from different elicitation methods. Ultimately, I think this is not a weakness of the current study, but rather of the field as a whole. I would suggest the authors mention this problem in the discussion and call for more specific cognitive models and theories that build stronger connections between elicitation methods, constructs, and cognitive processes influencing behavior.

References:

Frey, R., Pedroni, A., Mata, R., Rieskamp, J., & Hertwig, R. (2017). Risk preference shares the psychometric structure of major psychological traits. Science Advances, 3(10), e1701381.

Response 3: We agree with your comment and have added this reference and made the recommendation you outlined in the discussion section.

Example (p.30-31): Additionally, it is important to point that much of the literature that this paper builds on is based on fundamentally different experimental designs. As Frey et al. (2017) have already shown, behavioural elicitation methods for risk preferences often show low correlations between each other, making it not as straight-forward as presented previously to assume a similar underlying relation of different elicitation methods. While we argue that this is not primarily a weakness of the currently, we see it as general issue with the field overall, suggesting that more specific cognitive models and theories that build stronger connections between elicitation methods, constructs, and cognitive processes influencing behaviour would be sorely needed.

Reviewer #2

General Comment: I appreciate the authors effort to give more weight to Final Choice. The paper has been improved a lot for sure. However, I’m not sure whether the authors understand my major concern well. The point is that you have two moving parts in your Main Choice design. Such a design flaw makes it unable to test your hypotheses and hence is totally invalid. 

General Response: Thank you very much for giving us the opportunity to revise the paper further and for acknowledging that our previous revisions improved the paper a lot. We have now followed your recommendation in full and are controlling for the amount of donation in all analyses regarding Main Choice and have updated our equivalence test results and Bayesian analyses accordingly. 

We would also like to thank you for spelling out your main objection to our Main Choice analysis in much more detail in this round of review. You were right that we had not fully grasped it at the last R&R stage, but we now believe that we have a much better understanding of your main objection to this design. While we hope that you can appreciate that we cannot remove the Main Choice analyses from our paper due to them being our pre-registered hypotheses, we have made four significant changes to our paper that we hope will help address your point. First, we have put even more weight on Final Choice, for example by conducting Bayesian linear regression analyses with, reporting model-averaged coefficients for it (but not for Main Choice). Second, we have followed your recommendation and are now controlling for amount donated in all Main Choice analyses. Third, we now outline your full objection in detail in the discussion section, where we give it substantive space, outline your examples, and detail these challenges for the interpretation of data collected from Main Choice. This is to ensure that all readers are informed about your concerns regarding our design, without compromising on our commitment to conduct, report, and analyse our experiment as pre-registered. Fourth, we have conducted two sets of regressions (one in the appendix, one with same results in this letter) aimed at controlling for your concerns, showing no difference in results. We hope that this goes towards addressing your main concern, though if you have any further specific recommendations, we are happy to follow them too.

Comment 1: The reasoning is simple: when one can choose both the type of the charity and the amount to donate, one can choose two different compositions that one is indifferent from. For example, if I’m indifferent between $1/sure thing charity and $0.5/probabilistic charity, I can choose randomly from the two. Overall, you would find roughly 50% choose sure thing and 50% choose probabilistic charity. This means you cannot test the preference for charity type under such a design. This is point one. 

Response 1: Thank you very much for spelling out your main objection in more detail. We now believe that we better understand your main concern. Your concern is of course legitimate and true. In general, indifference is always a possibility (unless we make some assumptions in terms of the decision maker’s preferences). For example, even if we had given our subjects the choice between donating a fixed amount to the sure thing charity or to the probabilistic charity or donating nothing to either of them, a subject’s choice to donate to one of the two types of charities would not rule out the possibility that the decision maker was indifferent between donating the fixed amount to either of them. Of course, you are right that when the decision maker can choose not just between a sure thing charity and a probabilistic charity indifference but also the amount to be donated, then the indifference situation is more likely to arise than when the amount to be donated is fixed, and the only choice is to which charity to donate or to not donate at all. In the text, the new revised version now explicitly acknowledges the indifference issue you pointed out.

There is, however, a way to delve further into the data to get an idea of the concern you raised. That way involves considering a participant’s choices at Main Choice and Final Choice simultaneously to identify possible cases of indifference. We consider that those who: i) donated to a sure-thing charity in Main Choice but did not donate at all in Final Choice when they were shown a probabilistic charity, might have been indifferent between a strictly positive donation to a sure-thing charity and donating nothing to a probabilistic charity; ii) donated to a probabilistic charity in Main Choice but did not donate at all in Final Choice when they were shown a sure-thing charity, might have been indifferent between a strictly positive donation to a probabilistic charity and donating nothing to a sure-thing charity. Thus, in Appendix F we exclude such subjects from the regressions of Main Choice. The regression estimates replicate findings of the regressions in the main body of the text. See Example 1 below.

Of course, Appendix F’s regressions which are based on the above exclusions do not fully eliminate all the subjects who might have been indifferent between donating a strictly positive amount to one type of charity and a zero donation to the other type of charity. The reasons for this are that: i) subjects who donated to a sure-thing charity at Main Choice and were faced with the decision to donate a sure-thing charity at Final Choice might have been indifferent between donating a positive amount to a sure-thing charity and donating zero to a probabilistic charity at Main Choice, but since they were not faced with a probabilistic charity at Final Choice they did not have the chance to make choices to reveal that indifference; i) likewise, subjects who donated to a probabilistic charity at Main Choice and were faced with the decision to donate a probabilistic charity at Final Choice might have been indifferent between donating a positive amount to a probabilistic charity and donating zero to a sure-thing charity at Main Choice, but since they were not faced with a sure-thing charity at Final Choice they did not have the chance to make choices to reveal that indifference.

This led us to re-run the regressions focusing on different exclusion criteria. We ran the regressions on the subsample of subjects who faced a probabilistic charity at Final Choice, but among those excluded the ones who had donated a positive amount to a sure-thing charity. The excluded subjects made decisions consistent with the indifference concern you stated (i.e., donated a strictly positive amount to a sure-thing charity but made no donation when faced with a probabilistic charity). The non-excluded subjects made decisions that are not consistent with that possibility. The regression estimates broadly replicate findings of the regressions in the main body of the text for this selected sample of subjects. The results for this regression are below (but not in the paper) as the results are the same as in Appendix F. 

Table X — Review Response Letter Regression Results for Charitable Giving Behaviour in Main Choice

 Predicting Choice between Sure-Thing and Probabilistic Charities

 (X)

Risk Attitude .496 (.980)

Ambiguity Aversion .008 (.015)

Numeracy -.002 (.015)

Empathy -.047 (.081)

Optimism

 .004* (.007)

Donor Type 

 Warm-Glow -.014 (.008)

 Pure Altruism .002 (.120)

Donation (amount) -.082 (.155)

Age -.001 (.002)

Gender .006 (.005)

Education 

 Undergraduate degree .066 (.124)

 Postgraduate/Professional degree .215 (.129)

Religion 

 Protestantism .056** (.125)

 Catholicism -.350 (.152)

 Islam -.030 (.163)

 Judaism -.115 (.339)

 Buddhism -.257 (.538)

 Hinduism .307 (.584)

Religious Participation .190 (.203)

Marriage Status .118 (.120)

Children -.067 (.133)

Financial Wellbeing .030 (.067)

Employment 

 Out of the workforce -.347 (.229)

 Part-time employment .052 (.205)

 Full-time employment -.139 (.198)

Adj. R2 .009

Sample size 93

Notes: OLS regression reporting unstandardised coefficients and standard errors. Outcome variable is charity choice (0 = sure-thing charity, 1 = probabilistic charity). *p<.1, **p<.05, ***p<.01, ****p<.001

We very much hope that these additional analyses and regressions make headway to address the concerns you raised.

Furthermore, as outlined in General Response, we will now present your main objection in significant detail in the discussion section such that readers are fully aware of this design limitation and its impact on the results. We also outline your criticism throughout the paper by pointing out repeatedly that Final Choice ought to be given more weight. See Example 2 and Example 3 below.

Example 1 (p. 48-49):

In Appendix Table 11, Model (17), we report the same OLS regression for Main Choice as in Model (1). The main change in this regression is that, in response to the worry that by excluding those that do not donate, we may be biasing our estimates as participants may be indifferent between donating a non-zero amount to one type of charity (say a sure-thing charity) and no donation to a probabilistic charity. As we had additional data on all participants from their choices in Final Choice, we excluded the following participants on top of our standard exclusion criteria. First, we excluded those who donated to a sure-thing charity in Main Choice but did not donate at all in Final Choice when they were shown a probabilistic charity (n=70). Second, we also excluded those who donated to a probabilistic charity in Main Choice but who did not donate at all in Final Choice when they were shown a sure-thing charity (n=5). Overall, we do not find a difference in results which suggests that that our data may go some way towards addressing this worry.

APPENDIX TABLE 11—REGRESSION RESULTS FOR CHARITABLE GIVING BEHAVIOUR IN MAIN CHOICE

 PREDICTING CHOICE BETWEEN SURE-THING AND PROBABILISTIC CHARITIES

 (17)

Risk Attitude .000 (.007)

Ambiguity Aversion -.004 (.006)

Numeracy -.005 (.030)

Empathy -.001 (.003)

Optimism

 -.004 (.004)

Donor Type 

 Warm-Glow -.012 (.065)

 Pure Altruism -.086 (.076)

Donation (amount) -.001 (.001)

Age .004 (.002)

Gender -.002 (.062)

Education 

 Undergraduate degree .068 (.064)

 Postgraduate/Professional degree -.017 (.069)

Religion 

 Protestantism -.116 (.084)

 Catholicism -.045 (.089)

 Islam -.111 (.171)

 Judaism -.198 (.412)

 Buddhism -.151 (.318)

 Hinduism .319 (.303)

Religious Participation .057 (.109)

Marriage Status .044 (.062)

Children .000 (.069)

Financial Wellbeing .007 (.030)

Employment 

 Out of the workforce -.213 (.122)

 Part-time employment -.005 (.104)

 Full-time employment -.099 (.096)

R2 .081

Sample size 232

Notes: OLS regression reporting unstandardised coefficients and standard errors. Outcome variable is charity choice (0 = sure-thing charity, 1 = probabilistic charity). *p<.1, **p<.05, ***p<.01, ****p<.001

Example 2 (p. 31-32): Moreover, there is an additional reason to significantly favour the results of Final Choice over those of Main Choice throughout the interpretation of this paper’s data. This is because the design of Main Choice has some inherent flaws that make interpretation of results difficult. This is because when participants make choices between two charities, they can make choices between compositions that they are indifferent between. For example, any one donor may be indifferent between a donation of £0.25 to a sure-thing charity and a donation of £0.10 to a probabilistic charity. In this scenario, the donor would choose randomly between the two. This in itself makes it difficult to cleanly identify a preference for one charity type or the other in the design that we have pre-registered and analysed in this paper. Based on this reason, we argue that the results from Final Choice should be given greater weight compared to Main Choice throughout this paper.

This concern is even more pointed if one considers that it may be the case that donors could be indifferent between donating a positive amount to a sure-thing charity and no donation to a probabilistic charity. As before, if they are indifferent between the two and choose randomly, this inhibits straightforward interpretation of our results. This is because we exclude all participants who choose not to donate (to understand their behaviour between these two options). Further, because it is quite plausible that, given the distribution of donation choices (most people who donate donate to a sure-thing charity), we may be excluding significantly more people favouring probabilistic charities but not sure-thing charities from those who are indifferent between a non-zero donation to a sure-thing charity and a zero donation to a probabilistic (and thus choose randomly between the two). 

Some of these concerns have been addressed ex-post in this paper, for example by controlling for amount donated in the analyses of Main Choice that we did not initially pre-register. While this goes some way towards addressing this concern, we argue that some fundamental design constraints of the set-up that we chose remain. As such, we have put a higher emphasis on our analyses of Final Choice (which do not fall prey to the same structural challenges) throughout this paper and argue that one ought to be generally cautious in interpreting the results of Main Choice. However, given that it was pre-registered, we continue to report it fully (and where we deviate from the pre-registered protocol, we document this in detail and provide the original analyses in the appendix). We hope that this discussion, highlighting these issues in detail, properly contexualises the results for readers. 

Example 3 (p. 15): This condition [Final Choice] controls for a number of potential confounds in Main Choice which allows for it to answer the paper’s central question more directly and cleanly, though it being the last task of the experiment, we cannot rule out potential order effects.

Comment 2: Point two, suppose people are more likely to donate more under sure-thing charity, then when you excluded 0 donation as you preregistered, you exclude more people who chose probabilistic charity (you can test this with your data). Th reason is that there must be some people who are indifferent between $positive amount/sure thing charity and $0/probabilistic charity, and these people chose randomly between the two. In the observed results, you excluded those who chose probabilistic charity but not the other type. 

Response 2: Thank you again for further clarifications on this. As outlined above, we now discuss in detail your objection in the discussion section, including the examples you have provided here to ensure we do not misrepresent your main concern, while also offering regressions that hopefully go some way towards addressing this concern. Further, in order to make sure we are not missing something in our exchanges, we re-read our paper line by line. In doing so, we have realised that we have not made clear enough in the main paper that participant choices in Main Choice are between donating a non-zero amount to either of the two charities or not donating at all. Participants cannot choose a charity and then enter ‘0’. We have now made this clear in the paper at multiple points and apologise for this oversight in our previous revision. 

Example (p. 5, 9, 14, 29): In attempting the construction of an externally valid donation choice that allows us to capture the difference between sure-thing charities and probabilistic charities in actual charities that participants can donate money to, our main outcome variables of interest are (i) the choice between two real charities, one sure-thing charity and one probabilistic charity and (ii) the choice about one of those charities that has been randomly selected. Both of these charitable decision-making scenarios have strengths and weaknesses from an experimental design perspective, but we hope that they jointly allow us to better understand the role of individual differences in charitable decision-making scenarios like these. We outline the main weakness of (i) in the discussion section and argue that, overall, (ii) is a cleaner design. 

Lastly, one may also be interested in donation behaviour not between these two types of charities, but rather just in the context where potential donors are presented with one such charity. This may reduce the chance of additional confounds (like worrying that the design that presents two charities is artificial in its dichotomous presentation; after all, most naturalistic decisions are not decisions between two distinct choices). It also is overall a cleaner design that brings with it less drawbacks regarding interpretation of results.

In both cases, we collect data on whether they donate at all, if they do, which charity they choose in Main Choice (participants can choose up to one charity), and how much they donate. In other words, participants can donate a non-zero amount to either charity or not donate at all (i.e. one cannot select a charity and choose to donate ‘0’).

We have some evidence in favour of this worry as at least some of the pre-registered factors predict directionally as expected in Final Choice, where only one charity was presented to participants. As such, it may be justified to put more interpretative emphasis on the results from Final Choice compared to Main Choice.

Comment 3: Point 3, because of the above issue, if you regress charity type preference on other variables, you have to control for the amount donated, as an ex-post control, because you did not control for the donate amount in the experiment. I’m not clear what endogeneity this may cause. But if there is any, I don’t believe the data from Main Choice can prove anything. There is no problem if you want to report the analyses as you preregistered. But a preregistration doesn’t mean your design and analysis (Main Choice) is not problematic.

Response 3: Thank you very much for clarifying your comment from last time. As a result of your explanation of your main objection above, we have now followed your recommendation in full by controlling for the amount donated as an ex-post control for all analyses of Main Choice. 

Example (p. 20-22): First, we investigate general donation behaviour relating to Main Choice. The results presented in Table 3 speak to the central null hypotheses #1, #2, and #3. Model (1) reports the results for the Main Condition. The outcome variable is the type of charity conditional on a donation being made, with 0 being coded as the sure-thing charity and 1 as the probabilistic charity. The gender variable is coded 1 for female, all other categorical variables are coded 1 for the affirmative. As outlined above, the risk attitude measure is a discrete variable of the number of boxes opened, the ambiguity aversion is the result of the subtraction of the reservation prices. All other scales are the sum of the (re-reversed) individual items. As specified in our pre-registration, we report main regression results for binary outcomes using an OLS model and also for the corresponding logit model as a robustness check in Appendix B (Appendix Table 2) to check for sensitivity to functional form choice, where we find no impact of model choice. We also report a robustness check in Appendix B (Appendix Table 3) where we report a regression with random effects at the stimulus level, finding that our null result is also robust to this model choice. 

Table 3—Regression Results for Charitable Giving Behaviour in Main Choice

 Predicting Choice between Sure-Thing and Probabilistic Charities

 (1)

Risk Attitude .000 (.005)

Ambiguity Aversion -.003 (.005)

Numeracy .004 (.024)

Empathy <.001 (.003)

Optimism

 -.004 (.003)

Donor Type 

 Warm-Glow -.014 (.051)

 Pure Altruism -.062 (.064)

Donation (amount) -.001 (.001)

Age .003 (.002)

Gender .013 (.050)

Education 

 Undergraduate degree .044 (.051)

 Postgraduate/Professional degree -.020 (.055)

Religion 

 Protestantism -.082 (.067)

 Catholicism .012 (.073)

 Islam -.101 (.122)

 Judaism -.163 (.379)

 Buddhism -.105 (.287)

 Hinduism .378 (.276)

Religious Participation .041 (.089)

Marriage Status .045 (.050)

Children -.007 (.054)

Financial Wellbeing .005 (.024)

Employment 

 Out of the workforce -.154 (.097)

 Part-time employment .003 (.085)

 Full-time employment -.054 (.079)

R2 .064

Sample size 307

Notes: OLS regression reporting unstandardised coefficients and standard errors. Outcome variable is charity choice (0 = sure-thing charity, 1 = probabilistic charity). *p<.1, **p<.05, ***p<.01, ****p<.001

We find that none of the main variables nor the demographic control variables predict charity choice in the Main Condition. We can straightforwardly conclude from Model (1) that we do not have evidence to reject the null hypotheses #1 and #2 as none of the independent variables meaningfully predict donor behaviour. However, note that Model (1) includes the control variable of amount donated that was not pre-registered but suggested by a helpful anonymous reviewer. For the pre-registered regression model without this control with no difference in results, please see Appendix A.

---

## [Decision Letter · Decision Letter 2]

16 Aug 2022

PONE-D-21-35635R2Sure-Thing vs. Probabilistic Charitable Giving: Experimental Evidence On the Role of Individual Differences in Risky and Ambiguous Charitable Decision-MakingPLOS ONE

Dear Dr. Schoenegger,

Thank you for submitting your manuscript to PLOS ONE. After careful consideration, we feel that it has merit but does not fully meet PLOS ONE’s publication criteria as it currently stands. Therefore, we invite you to submit a revised version of the manuscript that addresses the points raised during the review process. Please explain the limitations about this kind of study using web-based sample. 

We look forward to receiving your revised manuscript.

Kind regards,

Junhuan Zhang, PhD

Academic Editor

PLOS ONE

Journal Requirements:

Reviewers' comments:

Reviewer's Responses to Questions

**Comments to the Author**

1. If the authors have adequately addressed your comments raised in a previous round of review and you feel that this manuscript is now acceptable for publication, you may indicate that here to bypass the “Comments to the Author” section, enter your conflict of interest statement in the “Confidential to Editor” section, and submit your "Accept" recommendation.

Reviewer #2: All comments have been addressed

2. Is the manuscript technically sound, and do the data support the conclusions?

Reviewer #2: (No Response)

3. Has the statistical analysis been performed appropriately and rigorously? 

Reviewer #2: (No Response)

4. Have the authors made all data underlying the findings in their manuscript fully available?

Reviewer #2: (No Response)

5. Is the manuscript presented in an intelligible fashion and written in standard English?

Reviewer #2: (No Response)

6. Review Comments to the Author

Reviewer #2: I thank the authors for their excellent second round revisions which I believe make the paper strong and publishable. It is smart to connect the data from the Main and Final treatments. It is now clear and important to point out the flaws in the Main treatment and putting more weight to the Final treatment. Good work!

7. PLOS authors have the option to publish the peer review history of their article (what does this mean?). If published, this will include your full peer review and any attached files.

Reviewer #2: No

---

## [Author Response · Author response to Decision Letter 2]

18 Aug 2022

Dear Editor,

Below we outline our response to your last comment. 

Comment: Please explain the limitations about this kind of study using web-based sample.

Response: We agree that we have not properly stated this limitation before and now do so explicitly in the revised manuscript. 

Example (p. 33): A second potential limitation is the use of a web-based sample from Prolific. While we have tried to counterbalance this concern by relying on a representative sample (at least along the dimensions of age, sex, and ethnicity), there may still be some dimension along which our sample is not representative of the population as a whole. This, in turn, means that that there may be some external validity worries inherent in using this sample, which may impact generalisability of our results.

---

## [Editor Report · Decision Letter 3]

19 Aug 2022

Sure-Thing vs. Probabilistic Charitable Giving: Experimental Evidence On the Role of Individual Differences in Risky and Ambiguous Charitable Decision-Making

PONE-D-21-35635R3

Dear Dr. Schoenegger,

We’re pleased to inform you that your manuscript has been judged scientifically suitable for publication and will be formally accepted for publication once it meets all outstanding technical requirements.

Kind regards,

Junhuan Zhang, PhD

Academic Editor

PLOS ONE
---

## [Editor Report · Acceptance letter]

4 Sep 2022

PONE-D-21-35635R3 

Sure-Thing vs. Probabilistic Charitable Giving: Experimental Evidence on the Role of Individual Differences in Risky and Ambiguous Charitable Decision-Making 

Dear Dr. Schoenegger:

I'm pleased to inform you that your manuscript has been deemed suitable for publication in PLOS ONE. Congratulations! Your manuscript is now with our production department. 

Kind regards, 

on behalf of

Dr. Junhuan Zhang 

Academic Editor

PLOS ONE